# Nested Variational Inference

**Heiko Zimmermann**[†]
h.zimmermann@uva.nl

**Hao Wu**[*]
wu.hao10@northeastern.edu

**Babak Esmaeili**[†]
b.esmaeili@uva.nl

**Jan-Willem van de Meent**[†*]
j.w.vandemeent@uva.nl

[†]Institute of Informatics, University of Amsterdam
[*]Khoury College of Computer Sciences, Northeastern University

## Abstract

We develop nested variational inference (NVI), a family of methods that learn proposals for nested importance samplers by minimizing an forward or reverse KL divergence at each level of nesting. NVI is applicable to many commonly-used importance sampling strategies and provides a mechanism for learning intermediate densities, which can serve as heuristics to guide the sampler. Our experiments apply NVI to (a) sample from a multimodal distribution using a learned annealing path (b) learn heuristics that approximate the likelihood of future observations in a hidden Markov model and (c) to perform amortized inference in hierarchical deep generative models. We observe that optimizing nested objectives leads to improved sample quality in terms of log average weight and effective sample size.

## 1 Introduction

Deep generative models provide a mechanism for incorporating priors into methods for unsupervised representation learning. This is particularly useful in settings where the prior defines an inductive bias that reflects the structure of the underlying domain. Training models with structured priors, however, poses some challenges. A standard strategy for training deep generative models is to maximize a reparameterized evidence lower bound with respect to both the generative and an inference model [Kingma, Welling, 2013; Rezende et al., 2014]. This approach works well in variational autoencoders with isotropic Gaussian priors, but often fails for models with more structured priors or likelihoods.

In recent years, a range of strategies for improving upon standard reparameterized variational inference have been put forward. These include wake-sleep style variational methods that minimize the forward KL-divergence [Bornschein, Bengio, 2015; Le et al., 2019] as well as sampling schemes that incorporate annealing [Huang et al., 2018], Sequential Monte Carlo [Le et al., 2018; Naesseth et al., 2017; Maddison et al., 2017], Gibbs sampling [Wu et al., 2019; Wang et al., 2018], and MCMC updates [Salimans et al., 2015; Hoffman, 2017; Li et al., 2017]. While these methods offer flexible inference, typically resulting in better approximations to the posterior compared to traditional variational inference methods, they are either model-specific, requiring specialized sampling schemes and gradient estimators, or can not be easily composed with other techniques.

In this paper, we propose nested variational inference, a framework for combining nested importance sampling and variational inference. Nested importance sampling formalizes the construction of proposals by way of calls to other importance samplers [Naesseth et al., 2015; Naesseth et al., 2019], Many existing importance samplers are instances of nested samplers, including methods based on annealed importance sampling [Neal, 2001] and sequential Monte Carlo [Del Moral et al., 2006].

35th Conference on Neural Information Processing Systems (NeurIPS 2021).

NVI learns proposals by optimizing a divergence at each level of nesting. Additionally, we combine nested variational objectives with importance resampling to further improve sampling quality, without the need to undergo extra steps to maintain differentiability due to the local nature of the objective. Resampling also allows to compute gradient estimates based on incremental weights, which depend only on variables that are sampled locally, rather than on all variables in the model. Doing so yields lower variance weights, whilst maintaining a high sample diversity relative to existing methods.

## 2 Background

### 2.1 Stochastic Variational Inference.

Stochastic variational methods approximate a target density $\pi(z; \theta) = \gamma(z; \theta)/Z$ with parameters $\theta$ using a variational density $q(z; \phi)$ with parameters $\phi$. Two common variational objectives are the *forward* and *reverse* Kullback-Leibler (KL) divergence, which are both instances of $f$-divergences

$$D_f\left(\pi \parallel q\right) = \mathop{\mathbb{E}}_q\left[f\left(\frac{\pi(z; \theta)}{q(z; \phi)}\right)\right], \tag{1}$$

with $f(w) = w \log w$ and $f(w) = -\log w$ respectively. We are typically interested in the setting where $\pi(z; \theta)$ is the posterior $p_\theta(z \mid x)$ of a model with latent variables $z$ and observations $x$. In this case, $\gamma(z; \theta) = p_\theta(x, z)$ is the joint density of the model, and $Z = p_\theta(x)$ is the marginal likelihood.

**Reverse KL-divergence.** When optimizing $\mathrm{KL}\left(q \parallel \pi\right)$, known as the reverse or *exclusive* KL divergence, it is common practice to maximize a lower bound $\mathcal{L} = \mathbb{E}_q[\log(\gamma/q)] = \log Z - \mathrm{KL}\left(q \parallel \pi\right)$ with respect to $\theta$ and $\phi$. The gradient of $\mathcal{L}$ can be approximated using reparameterized samples [Kingma, Welling, 2013; Rezende et al., 2014], likelihood-ratio estimators [Wingate, Weber, 2013; Ranganath et al., 2014], or a combination of the two [Schulman et al., 2015; Ritchie et al., 2016a].

**Forward KL-divergence.** In the case of the forward divergence $\mathrm{KL}\left(\pi \parallel q\right)$, also known as the *inclusive* KL divergence, stochastic variational methods typically optimize separate objectives for the inference and generative model. A common strategy is to train the generative model by either maximizing the ELBO or likelihood on the data. Optimizing the inference model requires samples from $\pi$, which itself requires approximate inference. A common strategy, which was popularized in the context of reweighted wake-sleep (RWS) methods [Bornschein, Bengio, 2015; Le et al., 2019], is to use $q$ as a proposal in an importance sampler.

### 2.2 Importance Sampling

**Self-Normalized Importance Samplers.** An expectation $\mathbb{E}_\pi[g(z)]$ with respect to $\pi$ can be rewritten with respect to a proposal density $q$ by introducing an unnormalized importance weight $w$,

$$\mathop{\mathbb{E}}_\pi\left[g(z)\right] = \frac{1}{Z} \mathop{\mathbb{E}}_q\left[w\, g(z)\right], \qquad\qquad w = \frac{\gamma(z; \theta)}{q(z; \phi)}. \tag{2}$$

Self-normalized estimators use weighted samples $(w^{(s)}, z^{(s)})_{s=1}^S$ to both approximate the expectation with respect to $q$, and to compute an estimate $\hat{Z}$ of the normalizing constant,

$$\mathop{\mathbb{E}}_\pi[g(z)] \simeq \hat{g} = \frac{1}{\hat{Z}} \frac{1}{S} \sum_{s=1}^S w^{(s)}\, g(z^{(s)}), \qquad \hat{Z} = \frac{1}{S} \sum_{s=1}^S w^{(s)}, \qquad z^{(s)} \sim q(\cdot; \phi). \tag{3}$$

The resulting estimator is consistent, i.e. $\hat{g} \xrightarrow{a.s.} \mathbb{E}_\pi[g(z)]$ as the number of samples $S$ increases , but it is not unbiased, since it follows from Jensen's inequality that $1/Z = 1/\mathbb{E}_q[\hat{Z}] \le \mathbb{E}_q[1/\hat{Z}]$ . If the discrepancy between $q$ and $\pi$ is large, the importance weights will have large variance, which also has an impact on the bias. When $q = \pi$, the importance weight $w = Z$ has zero variance, and the Jensen's inequality is tight. In the context of stochastic variational inference, this means that gradient estimates might initially be strongly biased, since there will typically be a large discrepancy between $q$ and $\pi$. However, the variance will typically decrease as the quality of the approximation improves.

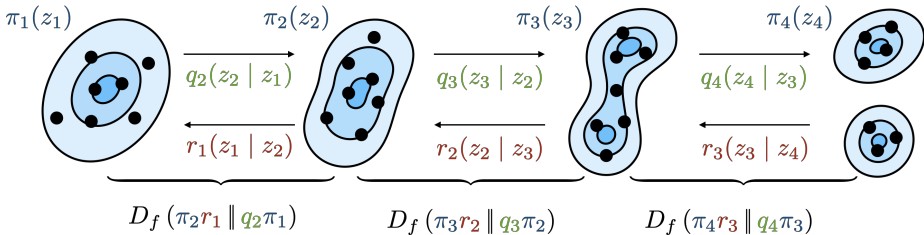

Figure 1: Nested variational inference minimizes an $f$-divergence at each step in a sequence of densities to learn forward proposals $q_k$, reverse kernels $r_{k-1}$, and intermediate densities $\pi_k$.

**Nested importance sampling and proper weighting**  Nested Importance sampling formalizes the construction of proposals by way of calls to other importance samplers. Extending the example above, we now assume $q$ is some unnormalized density itself that we cannot directly generates samples from. We can employ another importance sampler with proposal $\eta$ to simulate samples from $q$,

$$z^{(l)} \sim \eta(\cdot, \psi), \qquad w^{(l)} = \frac{q(z^{(l)}; \phi)}{\eta(z^{(l)}; \psi)}, \qquad \hat{Z}_q = \frac{1}{L} \sum_{l=1}^{L} w^{(l)}$$

The resulting weighted samples $(w^{(l)}, z^{(l)})_{l=1}^{L}$ can subsequently be used compute the normalizing constant $Z_q$ of $q$ and to generate samples that are approximately distributed w.r.t. $q$ via resampling. This allows us to use $q$ as a proposal in a subsequent importance sampling step, whose weighted samples can again be used to compute consistent estimates w.r.t. the target density of interest $\pi$. This can be generalized, by introducing the notion of proper weighting.

**Definition 2.1** (Proper weighting). *Let $\pi$ be a probability density. For some constant $c > 0$, a random pair $(w, z) \sim \Pi$ is properly weighted (p.w.) for an unnormalized probability density $\gamma \equiv Z\pi$ if $w \geq 0$ and for all measurable functions $g$ it holds that*

$$\mathbb{E}_{w,z\sim\Pi} [w\, g(z)] = c \int dz\, \gamma(z)\, g(z) = cZ \mathbb{E}_{z\sim\pi} [g(z)].$$

As long as we can generate properly weighted samples for the target density $\pi$ from some sampler $\Pi$, we can compute consistent estimates

$$\frac{\frac{1}{S} \sum_{s=1}^{S} w^{(s)}\, g(z^{(s)})}{\frac{1}{S} \sum_{s=1}^{S} w^{(s)}} = \frac{\cancel{c}\cancel{Z}\, \mathbb{E}_{z\sim\pi} [g(z)]}{\cancel{c}\cancel{Z}} = \mathbb{E}_{z\sim\pi} [g(z)]. \tag{4}$$

## 3  Nested Variational Inference

A widely used strategy in importance sampling is to decompose a difficult sampling problem into a series of easier problems. A common approach is to define a sequence of unnormalized densities $\{\gamma_k\}_{k=1}^{K}$ that interpolate between an initial density $\pi_1 = \gamma_1/Z_1$, for which sampling is easy, and the final target density $\pi_K = \gamma_K/Z_K$. At each step, samples from the preceding density serve to construct proposals for the next density, which is typically combined with importance resampling or application of a Markov chain Monte Carlo (MCMC) operator to improve the average sample quality.

**Nested variational objectives.** NVI defines objectives for optimizing importance samplers that target a sequence of densities. At every step, it minimizes the discrepancy between a *forward density* $\hat{\pi}_k = \hat{\gamma}_k/Z_{k-1}$, which acts as the proposal, and a *reverse density* $\check{\pi}_k = \check{\gamma}_k/Z_k$, which defines an intermediate target. We define the forward density by combining the preceding target $\gamma_{k-1}$ with a forward kernel $q_k$, and the reverse density by combining the next target $\gamma_k$ with a reverse kernel $r_{k-1}$,

$$\check{\gamma}_k(z_k, z_{k-1}) = \gamma_k(z_k)\, r_{k-1}(z_{k-1}|z_k), \qquad \hat{\gamma}_k(z_k, z_{k-1}) = q_k(z_k|z_{k-1})\, \gamma_{k-1}(z_{k-1}). \tag{5}$$

Our goal is to learn pairs of densities $\check{\pi}_k$ and $\hat{\pi}_k$ that are as similar as possible. To do so, we minimize a variational objective $\mathcal{D}$ that comprises an $f$-divergence for each step in the sequence, along with a divergence between the first intermediate target $\pi_1$ and an initial proposal $q_1$. Concretely, the NVI objective can be formulated as,

$$\mathcal{D}_{\text{NVI}} = D_f\big(\pi_1 \,\big|\big|\, q_1\big) + \sum_{k=2}^{K} D_f\big(\check{\pi}_k \,\big|\big|\, \hat{\pi}_k\big). \tag{6}$$

Since each intermediate density $\pi_k$ occurs in both $\check{\pi}_k$ and in $\hat{\pi}_{k+1}$, this defines a trade-off between maximizing the similarity to $\pi_{k+1}$ and the similarity to $\pi_{k-1}$. To optimize this objective, we need to be able to simulate samples from the intermediate densities.

**Sampling from intermediate densities.** Given a pair $(w_{k-1}, z_{k-1})$ that is properly weighted for the previous target density $\gamma_{k-1}$, we can use a sequential Monte Carlo sampling [Del Moral et al., 2006] construction to define a pair $(w_k, z_k)$ that is properly weighted for $\gamma_k$,

$$z_k \sim q_k(\cdot \mid z_{k-1}), \qquad w_k = v_k\, w_{k-1}, \qquad v_k = \frac{\check{\gamma}_k(z_k, z_{k-1})}{\hat{\gamma}_k(z_k, z_{k-1})}. \qquad (7)$$

We refer to the ratio $v_k$ as the incremental weight. In this construction, $(w_k, z_{k-1:k})$ is properly weighted for $\check{\gamma}_k$ which implies that $(w_k, z_k)$ is also properly weighted for $\gamma_k$, since

$$\int dz_{k-1}\, \check{\gamma}_k(z_k, z_{k-1}) = \int dz_{k-1}\, \gamma_k(z_k) r_{k-1}(z_{k-1} \mid z_k) = \gamma_k(z_k). \qquad (8)$$

Sequential importance sampling can be combined with other operations that preserve proper weighting, including rejection sampling, application of an MCMC transition operator, and importance resampling. This defines a class of samplers that admits many popular methods as special cases, including sequential Monte Carlo (SMC) [Doucet et al., 2001], annealed importance sampling (AIS) [Neal, 2001], and SMC samplers [Del Moral et al., 2006]. In this work we only consider variational transition kernels and importance resampling. These samplers vary in the sequences of densities they define. For example, AIS and SMC samplers both define intermediate density over a fixed common domain. Here, a common strategy is to define an annealing path $\gamma_k(z_k) = \gamma_1(z_k)^{1-\beta_k} \gamma_K(z_k)^{\beta_k}$ for $0 = \beta_1 < \beta_2 < \ldots < \beta_K = 1$. In contrast, when using SMC for state-space models we can define the intermediate target $\pi_k$ to be the the filtering distribution on the first $k$ states. In this setting, the dimensionality of the support increases at each step and we can omit the reverse kernel $r_{k-1}$. We are exploring both of these cases in Section 4.

### 3.1 Computing Gradient Estimates

The NVI objective can be optimized with respect to three sets of densities. We will use $\theta_k$, $\hat{\phi}_k$, and $\check{\phi}_k$ to denote the parameters of the densities $\pi_k$, $q_k$, and $r_k$ respectively. For notational convenience, we use $\check{\rho}_k = \{\theta_k, \check{\phi}_{k-1}\}$ to refer to the parameters of the reverse density $\check{\pi}_k$, and $\hat{\rho}_k = \{\hat{\phi}_k, \theta_{k-1}\}$ to refer to the parameters of the forward density $\hat{\pi}_k$,

**Gradients of the Forward KL divergence.** When we employ the forward KL as the objective, the derivative with respect to $\hat{\rho}_k$ can be expressed as (see Appendix E.3),

$$-\frac{\partial}{\partial \hat{\rho}_k} \mathrm{KL}\left(\check{\pi}_k \,\|\, \hat{\pi}_k\right) = \underset{\check{\pi}_k}{\mathbb{E}}\left[\frac{\partial}{\partial \hat{\rho}_k} \log \hat{\gamma}_k(z_k, z_{k-1}; \hat{\rho}_k)\right] - \underset{\pi_{k-1}}{\mathbb{E}}\left[\frac{\partial}{\partial \hat{\rho}_k} \log \gamma_{k-1}(z_{k-1}; \theta_{k-1})\right]. \qquad (9)$$

This case is the nested analogue of RWS-style variational inference. We can move the derivative into the expectation, since $\check{\pi}_k$ does not depend on $\hat{\rho}_k$. We then decompose $\log \hat{\pi}_k = \log \hat{\gamma}_k - \log Z_{k-1}$ and use the identity from Equation 21 to express the gradient $\log Z_{k-1}$ as an expectation with respect to $\pi_{k-1}$. The resulting expectations can be approximated using self-normalized estimators based on the outgoing weights $w_k$ and incoming weights $w_{k-1}$ respectively.

The gradient of the forward KL with respect to $\check{\rho}_k$ is more difficult to approximate, since the expectation is computed with respect to $\check{\pi}_k$, which depends on the parameters $\check{\rho}_k$. The gradient of this expectation has the form (see Appendix E.3)

$$-\frac{\partial}{\partial \check{\rho}_k} \mathrm{KL}\left(\check{\pi}_k \,\|\, \hat{\pi}_k\right) = -\underset{\check{\pi}_k}{\mathbb{E}}\left[\log v_k \frac{\partial}{\partial \check{\rho}_k} \log \check{\pi}_k(z_k, z_{k-1}; \check{\rho}_k)\right] \qquad (10)$$

$$= -\underset{\check{\pi}_k}{\mathbb{E}}\left[\log v_k \frac{\partial}{\partial \check{\rho}_k} \log \check{\gamma}_k(z_k, z_{k-1}; \check{\rho}_k)\right] + \underset{\check{\pi}_k}{\mathbb{E}}\left[\log v_k\right] \underset{\pi_k}{\mathbb{E}}\left[\frac{\partial}{\partial \check{\rho}_k} \log \gamma_k(z_k; \theta_k)\right].$$

In principle, we can approximate this gradient using self-normalized estimators based on the outgoing weight $w_k$. We experimented extensively with this estimator, but unfortunately we found it to be unstable, particularly for the gradient with respect to the parameters of the reverse kernel $r_{k-1}$. For this reason, our experiments employ the reverse KL when learning reverse kernels.

Our hypothesis is that the instability in this estimator arises because the gradient *decreases* the probability of high-weight samples and *increases* the probability of low-weight samples, rather than the other way around. This could lead to problems during early stages of training, when the estimator will underrepresent low-weight samples, for which the probability should increase.

**Gradients of the Reverse KL divergence.** When computing the gradient of the reverse KL with respect to $\hat{\rho}_k$, we obtain the nested analogue of methods that maximize a lower bound. Here we can either use reparameterized samples [Kingma, Welling, 2013; Rezende et al., 2014] or likelihood-ratio estimators [Wingate, Weber, 2013; Ranganath et al., 2014]. We will follow Ritchie et al. (2016b) and define a unified estimator in which proposals are generated using a construction

$$w_k = v_k w_{k-1}, \quad z_k = g(\tilde{z}_k, \hat{\phi}_k), \quad \tilde{z}_k \sim \tilde{q}_k(\tilde{z}_k \mid z_{k-1}, \hat{\phi}_k), \quad w_{k-1}, z_{k-1} \sim \Pi_{k-1}. \quad (11)$$

This construction recovers reparameterized samplers in the special case when $\tilde{q}_k$ does not depend on parameters, and recovers non-reparameterized samplers when $z_k = \tilde{z}_k$. This means it is applicable to models with continuous variables, discrete variables, or a combination of the two. The gradient of the reverse KL for proposals that are constructed in this manner becomes (see Appendix E.2)

$$
\begin{aligned}
-\frac{\partial}{\partial \hat{\rho}_k} \mathrm{KL}\left(\hat{\pi}_k \,\|\, \check{\pi}_k\right) = & \,\mathbb{E}_{\hat{\pi}_k}\left[\frac{\partial}{\partial z_k} \log \check{\gamma}_k\big(z_k, z_{k-1}; \hat{\rho}_k\big)\frac{\partial z_k}{\partial \hat{\rho}_k}\right] \\
& + \mathbb{E}_{\hat{\pi}_k}\left[\log v_k \frac{\partial}{\partial \hat{\rho}_k} \log \hat{\gamma}_k\big(z_k, z_{k-1}; \hat{\rho}_k\big)\right] \\
& - \mathbb{E}_{\hat{\pi}_k}\left[\log v_k\right]\mathbb{E}_{\pi_{k-1}}\left[\frac{\partial}{\partial \hat{\rho}_k} \log \gamma_{k-1}\big(z_{k-1}; \theta_{k-1}\big)\right].
\end{aligned}
\quad (12)
$$

In this gradient, the first term represents the pathwise derivative with respect to reparameterized samples. The second term defines a likelihood-ratio estimator in terms of the unnormalized density $\hat{\gamma}_k$, and the third term computes the contribution of the gradient of the log normalizer $\log Z_{k-1}$.

Computing the gradient of the reverse KL with respect to $\check{\rho}_k$ is once again straightforward, since we are computing an expectation with respect to $\hat{\pi}_k$, which does not depend on $\check{\rho}_k$. This means we can move the derivative into the expectation, which yields a gradient analogous to that in Equation 9,

$$-\frac{\partial}{\partial \check{\rho}_k} \mathrm{KL}\left(\hat{\pi}_k \,\|\, \check{\pi}_k\right) = \mathbb{E}_{\hat{\pi}_k}\left[\frac{\partial}{\partial \check{\rho}_k} \log \check{\gamma}_k\big(z_k, z_{k-1}; \check{\rho}_k\big)\right] - \mathbb{E}_{\pi_k}\left[\frac{\partial}{\partial \check{\rho}_k} \log \gamma_k\big(z_k; \theta_k\big)\right]. \quad (13)$$

**Variance Reduction.** To reduce the variance of the gradient estimates we use the expected log-incremental weight as a baseline for score function terms and employ the sticking-the-landing trick [Roeder et al., 2017] when reparameterizing the forward kernel as described in Appendix E.

## 3.2 Relationship to Importance-Weighted and Self-Normalized Estimators

There exists a large body of work on methods that combine variational inference with MCMC and importance sampling. We refer to Appendix A for a comprehensive discussion of related and indirectly related approaches. To position NVI in the context of the most directly related work, we here focus on commonly used importance-weighted and self-normalized estimators.

NVI differs from existing methods in that it defines an objective for pairs of variables $(z_k, z_{k-1})$ at each level of nesting, rather than a single objective for the entire sequence of variables $(z_1, \ldots, z_K)$. A standard approach for combining importance sampling and variational inference is to define an "importance-weighted" lower bound $\hat{\mathcal{L}}_K = \log \hat{Z}_K$ [Burda et al., 2016]. By Jensen's inequality, $\mathbb{E}[\hat{\mathcal{L}}_K] \leq \log \mathbb{E}[\hat{Z}_K] = \log Z_K$, which implies that we can define a stochastic lower bound using any properly-weighted sampler for $\gamma_K$, including samplers based on SMC [Le et al., 2018; Naesseth et al., 2018; Maddison et al., 2017]. For purposes of learning the target density $\gamma_K$, this approach is equivalent to computing an RWS-style estimator of the gradient,

$$\frac{\partial}{\partial \theta_K} \hat{\mathcal{L}}_K = \frac{1}{\hat{Z}_K}\frac{1}{S}\sum_{s=1}^{S} w_K^s \frac{\partial}{\partial \theta_K} \log \gamma_K(z_K^s; \theta_K). \quad (14)$$

However, these two approaches are not equivalent for purposes of learning the proposals. We can maximize a stochastic lower bound to learn $q_k$, but this requires doubly-reparameterized estimators

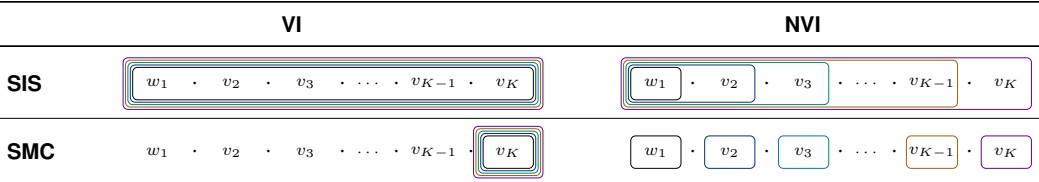

|  | **VI** | **NVI** |
|---|---|---|
| **SIS** | $w_1 \cdot v_2 \cdot v_3 \cdot \cdots \cdot v_{K-1} \cdot v_K$ | $w_1 \cdot v_2 \cdot v_3 \cdot \cdots \cdot v_{K-1} \cdot v_K$ |
| **SMC** | $w_1 \cdot v_2 \cdot v_3 \cdot \cdots \cdot v_{K-1} \cdot v_K$ | $w_1 \cdot v_2 \cdot v_3 \cdot \cdots \cdot v_{K-1} \cdot v_K$ |

Figure 2: Weight contributions in the self-normalized gradient estimators for the forward KL-divergence for VI and NVI using SIS (no resampling) and SMC (resampling). VI computes gradient estimates using the final weights (SIS), which simplify to the final incremental weight when resampling is performed (SMC). NVI computes gradient estimates based on the intermediate weights (SIS), which simplify to the intermediate incremental weights when resampling is performed (SMC).

[Tucker et al., 2018] in order to avoid problems with the signal-to-noise ratio in this estimator, which can paradoxically deteriorate with the number of samples [Rainforth et al., 2018]. The estimators in NVI do not suffer from this problem, since we do not compute the logarithm of an average weight.

NVI is also not equivalent to learning proposals with RWS-style estimators. If we use sequential importance sampling (SIS) to generate samples, a self-normalized gradient for the parameters of $q_k$ that is analogous to the one in Equation 24 has the form

$$\mathbb{E}_{\pi_K, r_{K-1}, \dots, r_1} \left[ \frac{\partial}{\partial \hat{\phi}_k} \log q_k(z_k \mid z_{k-1} ; \hat{\phi}_k) \right] \simeq \frac{1}{\hat{Z}_K} \frac{1}{S} \sum_{s=1}^{S} w_K^s \frac{\partial}{\partial \hat{\phi}_k} \log q_k(z_k^s \mid z_{k-1}^s ; \hat{\phi}_k). \quad (15)$$

Note that this expression depends on the final weight $w_K$. By contrast, a NVI objective based on the forward KL yields a self-normalized estimator that is defined in terms of the intermediate weight $w_k$

$$\mathbb{E}_{\pi_k, r_{k-1}} \left[ \frac{\partial}{\partial \hat{\phi}_k} \log q_k(z_k \mid z_{k-1} ; \hat{\phi}_k) \right] \simeq \frac{1}{\hat{Z}_k} \frac{1}{S} \sum_{s=1}^{S} w_k^s \frac{\partial}{\partial \hat{\phi}_k} \log q_k(z_k^s \mid z_{k-1}^s ; \hat{\phi}_k). \quad (16)$$

If instead of SIS we employ sequential importance resampling (i.e. SMC), then the incoming weight $w_{k-1}$ is identical for all samples. This means that we can express this estimator in terms of the incremental weight $v_k$ rather than the intermediate weight $w_k$

$$\mathbb{E}_{\pi_k, r_{k-1}} \left[ \frac{\partial}{\partial \hat{\phi}_k} \log q_k(z_k \mid z_{k-1} ; \hat{\phi}_k) \right] \simeq \sum_{s=1}^{S} \frac{v_k^s}{\sum_{s'=1}^{S} v_k^{s'}} \frac{\partial}{\partial \hat{\phi}_k} \log q_k(z_k^s \mid z_{k-1}^s ; \hat{\phi}_k). \quad (17)$$

We see that NVI allows us to compute gradient estimates that are localized to a specific level of the sampler. In practice, this can lead to lower-variance gradient estimates.

Having localized gradient computations also offers potential memory advantages. Existing methods typically perform reverse-mode automatic differentiation on an objective that is computed from the final weights (e.g. the stochastic lower bound). This means that memory requirements scale as $\mathcal{O}(SK)$ since the system needs to keep the entire computation graph in memory. In NVI, gradient estimates at level $k$ do not require differentiation of the incoming weights $w_{k-1}$, This means that it is possible to perform automatic differentiation on a locally-defined objective before proceeding to the next level of nesting, which means that memory requirements would scale as $\mathcal{O}(S)$. It should therefore in principle be possible to employ a large number of levels of nesting $K$ in NVI, although we do not evaluate the stability of NVI at large $K$ in our experiments.

## 4 Experiments

We evaluate NVI on three tasks, (1) learning to sample form an unnormalized target density where intermediate densities are generated using annealing, (2) learning heuristic factors to approximate the marginal likelihood of future observations in state-space models, and finally (3) inferring distributions over classes from small numbers of examples in deep generative Bayesian mixtures.

### 4.1 Sampling from Multimodal Densities via Annealing

A common strategy when sampling from densities with multiple isolated modes is to anneal from an initial density $\gamma_1$, which is typically a unimodal distribution that we can sample from easily, to the

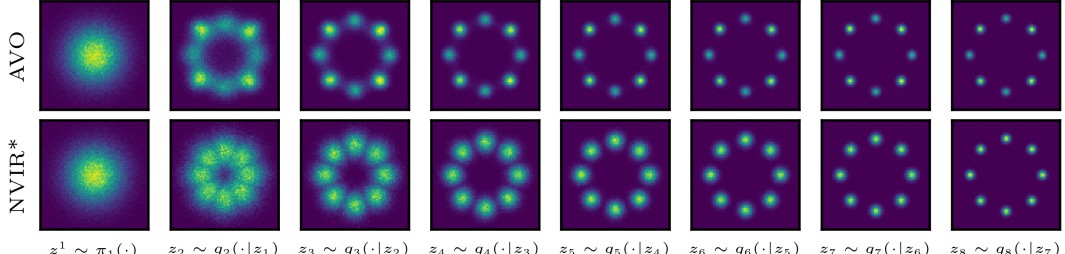

Figure 3: Samples from forward kernels trained with AVO, and NVIR$^*$.

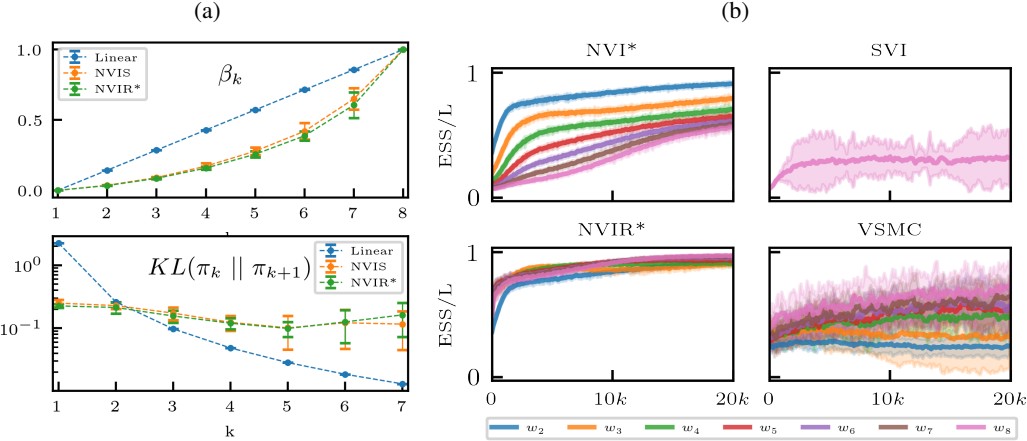

Figure 4: (a) ESS relative to the number of samples $L = 36$ during training for different methods using 7 pairs of transition kernels (sequence length $K = 8$) averaged across 10 independent runs. Error bars indicate $\pm 2$ standard deviations; mean and standard deviation are computed based a rolling average with window size 100. (b) (*Top*) Annealing paths learned by NVI$^*$ and NIVR* and the linearly spaced geometric annealing schedule (Linear) used by AVO, NVI, and NVIR. Results are averaged over 10 restarts; error bars indicate two standard deviations. (*Bottom*) Numerically computed KL-divergences between consecutive intermediate distributions for different schedules.

target density $\gamma_K$, which is multimodal [Neal, 2001]. Recent work on annealed variational objectives (AVOs) learns forward kernels $q_k$ and reverse kernels $r_{k-1}$ for an annealing sequence by optimizing a variational lower bound at each level of nesting [Huang et al., 2018].

$$\max_{q_k, r_k} \mathcal{L}_k^{\text{AVO}}, \quad \mathcal{L}_k^{\text{AVO}} = \mathbb{E}_{q_1, \dots, q_k} \big[ \log v_k \big], \quad \gamma_k(z) = \pi_1(z)^{1-\beta_k} \gamma_K(z)^{\beta_k}, \quad k = 1, \dots, K. \quad (18)$$

NVI allows us to improve upon AVO in two ways. First, we can perform importance resampling at every step to optimize an SMC sampler rather than an annealed importance sampler. Second, we can learn the annealing schedule $(\beta_1, \dots, \beta_K)$ as part of the intermediate densities $\gamma_k$ such that intermediate densities are scheduled more equidistantly in terms of KL-divergence.

We illustrate the effect of these two modifications in Figure 3, in which we compare AVO to NVI with resampling and a learned path, which we refer to as NVIR$^*$. Both methods minimize the reverse KL at each step [1]. The learned annealing path in NVIR$^*$ results in a *smoother* interpolation between the initial and final density. We also see that AVO does not assign equal mass to all 8 modes, whereas NVIR$^*$ yields a more even distribution.

In Figure 4a we compute the reverse KL between targets at each step in the sequence. For the standard linear annealing path, the KL decreases with $k$, suggesting that later intermediate densities are increasingly redundant. By constrast, in NVIR$^*$ we see that the KL is approximately constant across the path, which is what we would expect when minimizing $\mathcal{D}$ with respect to $\beta_k$. This is also the case in an ablation without resampling, which we refer to as NVI$^*$.

---

[1] As noted in Section 3.1, we found optimization of the forward KL to be unstable when learning $r_k$.

| | **log $\hat{Z}$** | | ($\log Z \approx 2.08$) | | **ESS** | | | |
|---|---|---|---|---|---|---|---|---|
| Seq. length | K=2 | K=4 | K=6 | K=8 | K=2 | K=4 | K=6 | K=8 |
| SVI | 1.86 | 1.89 | 1.92 | 1.72 | 51 | 47 | 32 | 25 |
| SVI-flow | 2.06 | - | - | - | 55 | - | - | - |
| AVO | 1.86 | 1.96 | 2.01 | 2.05 | 51 | 44 | 46 | 46 |
| NVI | 1.86 | 1.97 | 2.03 | 2.06 | 51 | 45 | 45 | 41 |
| NVIR | 1.86 | 1.98 | 2.04 | 2.06 | 51 | **99** | **98** | **97** |
| NVI* | 1.86 | **2.06** | **2.07** | 2.07 | 51 | 51 | 54 | 54 |
| NVIR* | 1.86 | **2.06** | **2.07** | **2.08** | 51 | 94 | 97 | 97 |
| AVO-flow | 2.05 | 2.07 | 2.07 | **2.08** | 28 | 66 | 76 | **70** |
| NVI*-flow | 2.05 | **2.08** | **2.08** | **2.08** | 28 | **81** | **79** | **70** |

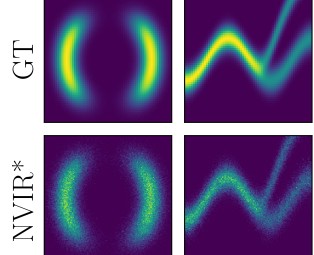

Table 1: Sample efficiency for NVI variants and baselines for $K-1$ annealing steps and $L$ samples per step for a fixed budget of $K \cdot L = 288$ samples. Metrics are computed for 100 batches of 100 samples per model across 10 restarts.

Figure 5: Ground truth densities (GT) and samples from final target density for NVIR* with 2 intermediate densities (K=4).

Figure 4b shows a rolling average of the ESS and its variance during training. We compare NVI-based methods to SVI and a variational SMC sampler [Le et al., 2018; Maddison et al., 2017; Naesseth et al., 2017]. NVIR* has consistently higher ESS and significantly lower variance compared to baselines. These plots also provides insight into the role of resampling in training dynamics. In NVI*, we observe a cascading convergence, which is absent in NVIR*. We hypothesize that resampling reduces the reliance on high-quality proposals from step $k-1$ when estimating gradients at step $k$.

Annealed NVI has similar use cases as normalizing flows [Rezende, Mohamed, 2015]. Inspired by concurrent work of Arbel et al. (2021), which explores a similar combination of SMC samplers and normalizing flows, we compare flow-based versions of NVI to planar normalizing flows, which maximize a standard lower bound (SVI-flow). We find that a normalizing flows can be effectively trained with NVI, in that samplers produce better estimates of the normalizing constant and higher ESS compared to SVI-flow. We also find that flow based models are able to produce high-quality samples with fewer intermediate densities (Figure 8). Moreover, we see that combining a flow-based proposal with learned $\beta_k$ values (NVI*-flow) results in a more accurate approximation of the target than in an ablation with a linear interpolation path (AVO-flow)[2].

In Table 1 we report sample quality in terms of the stochastic lower bound $\hat{\mathcal{L}}_K = \log \hat{Z}_K$ and the effective sample size ESS $= (\sum_s w_K^s)^2 / \sum_s (w_K^s)^2$. The first metric can be interpreted as a measure of the average sample quality, whereas the second metric quantifies sample diversity. We compare NVI with and without resampling (NVIR* and its NVI*) to ablations with a linear annealing path (NVIR and NVI), AVO, and a standard SVI baseline in which there are no intermediate densities. We additionally compare against AVO-flow and NVI*-flow, which employ flows. We observe that NVIR* and NVI*-flow outperform ablations and baselines in terms of $\log \hat{Z}$, and are competitive in terms of ESS. We show qualitative results for two additional target densities in Figure 5. For more details we refer to Appendix G.1.

### 4.2 Learning Heuristic Factors for State-space Models

Sequential Monte Carlo methods are commonly used in state-space models to generate samples by proposing one variable at a time. To do so, they define a sequence of densities $\pi_k = \gamma_k / Z_k$ on the first $k$ time points in a model, which are also known as filtering distributions,

$$\gamma_k(z_{1:k}, \eta) = p(x_{1:k}, z_{1:k}, \eta) = p(\eta) \, p(x_1, z_1 \mid \eta) \prod_{l=2}^{k} p(x_l, z_l \mid z_{l-1}, \eta), \quad Z_k = p(x_{1:k}). \quad (19)$$

Here $z_{1:k}$ and $x_{1:k}$ are sequences of hidden states and observations, and $\eta$ is a set of global variables of the model. The densities $\gamma_k$ differ from those in the annealing task in Section 4.1 in that the dimensionality of the support increases at each step. We define a forward density $\hat{\gamma}_k$ that combines the preceding target $\gamma_{k-1}$ with a proposal $q_k$ for the time point $z_k$, and define a reverse density $\check{\gamma}_k = \gamma_k$ that is equal to the next intermediate density (which means that we omit $r_{k-1}$),

$$\check{\gamma}_k(z_{1:k}, \eta) = \gamma_k(z_{1:k}, \eta), \qquad \hat{\gamma}_k(z_{1:k}, \eta) = q_k(z_k \mid x_k, z_{1:k-1}, \eta) \, \gamma_{k-1}(z_{1:k-1}, \eta). \quad (20)$$

A limitation of this construction is that the filtering distribution $\pi_{k-1}$ is not always a good proposal, since it does not incorporate knowledge of future observations. Ideally, we would like to define

---

[2]AVO-flow is itself a novel combination of AVO and flows, albeit an incremental one.

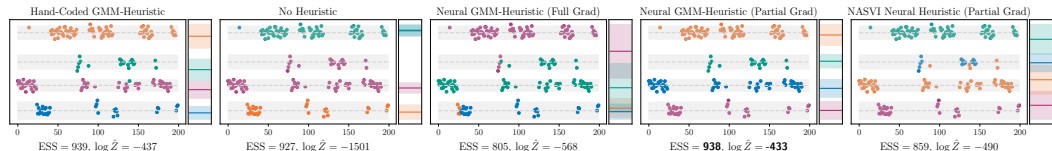

| Hand-Coded GMM-Heuristic | No Heuristic | Neural GMM-Heuristic (Full Grad) | Neural GMM-Heuristic (Partial Grad) | NASVI Neural Heuristic (Partial Grad) |
|---|---|---|---|---|
| ESS = 939, $\log \hat{Z} = -437$ | ESS = 927, $\log \hat{Z} = -1501$ | ESS = 805, $\log \hat{Z} = -568$ | ESS = **938**, $\log \hat{Z} = $ **-433** | ESS = 859, $\log \hat{Z} = -490$ |

Figure 6: (*Top*) qualitative results of an instance with $K = 200$ time steps (x-axis). Observations are color-coded based on the inferred assignments. Each colored band corresponds to the inferred cluster mean and standard deviation; grey bands indicate the ground truth of the clusters. (*Bottom*) We compute $\log \hat{Z}$ and the ESS using 1000 samples and report average values over 2000 test instances.

intermediate densities $\gamma_k(z_{1:k}, \eta) = p(x_{1:K}, z_{1:k}, \eta)$ that correspond to the smoothing distribution, but this requires computation of the marginal likelihood of future observations $p(x_{k+1:K} \,|\, z_k, \eta)$, which is intractable. This is particularly problematic when sampling $\eta$ as part of the SMC construction. The first density $\pi_1(z_1, \eta) = p(z_1, \eta \,|\, x_1)$ will be similar to the prior, which will result in poor sample efficiency, since the smoothing distribution $p(z_1, \eta \,|\, x_{1:K})$ will typically be much more concentrated.

To overcome this problem, we will use NVI to learn heuristic factors $\psi_\theta$ that approximate the marginal likelihood of future observations. We define a sequence of densities $(\gamma_0, \dots, \gamma_K)$,

$$\gamma_0(\eta) = p(\eta) \, \psi_\theta(x_{1:K}|\eta), \quad \gamma_k(z_{1:k}, \eta) = p(x_{1:k}, z_{1:k}, \eta) \, \psi_\theta(x_{k+1:K} \,|\, \eta), \quad k = 1, 2, ..., K.$$

Our goal is to learn parameters $\theta$ of the heuristic factor such that that intermediate densities approximate the smoothing distribution. This approach is similar to recently proposed work on twisted variational SMC [Lawson et al., 2018], which maximized a stochastic lower bound.

To evaluate the this approach, we will learn heuristic factors for a hidden Markov model (HMM). While HMMs are a well-understood model class, they are a good test case, in that they give rise to significant sample degeneracy in SMC and allow to compute tractable heuristics. We optimize an NVI objective based on the forward KL with respect to the heuristic factor $\psi_\theta$, an initial proposal $q_0(\eta \,|\, x_{1:K}; \phi)$ and a forward kernel $q_k(z_k \,|\, x_k, z_{k-1}, \eta; \phi_k)$. Figure 6 shows qualitative and quantitative results. We compare NVIR* with a neural GMM-heuristic to a baseline without a heuristic and a baseline that uses a hand-coded GMM-heuristic. In addition, we train models which employ the convex update operator proposed in the Automatic Structured Variational Inference (ASVI) [Ambrogioni et al., 2021]. Training ASVI by optimizing the ELBO failed due to the high variance of the score function estimator, which is needed for the discrete latent variables. However, we were able to train ASVI-based models with NVIR*. We refer to this novel combination as NASVI. Because NVI let us treat each step as a separate optimization problem, we also compare partial optimization with respect to $\hat{\gamma}_k$ only to full optimization with respect to both $\hat{\gamma}_k$ and $\check{\gamma}_k$. While full optimization yields poor results, partial optimization learns a neural heuristic whose performance is similar to the GMM heuristic, which is a strong baseline in this context. For more details see Appendix G.2.

### 4.3 Meta Learning with Deep Generative models

In this experiment, we evaluate NVI in the context of deep generative models with hierarchically-structured priors. Concretely, we consider the task of inferring class weights from a mini-batch of images in a fully unsupervised manner. For this purpose, we employ a variational autoencoder with a prior in the form of a Bayesian Gaussian mixture model. Unlike the previous experiments, here we both train a generative and an inference model. The forward kernels in this setting act as the *generative* model and the reverse kernels are the *inference* model. Moreover, in this model sampling simplifies because we design the intermediate densities to be tractable, hence no importance sampling construction is needed. Regardless, we still can construct an NVI objective to train hierarchical deep generative models more efficiently and with a more accurate posterior inference.

**Model Description.** We define a hierarchical deep generative model for batches of $N$ images of the form (see Appendix G.3 for a graphical model and architecture description)

$$\lambda \sim \text{Dir}(\cdot \,; \alpha) \quad c_n \sim \text{Cat}(\cdot \,|\, \lambda) \quad z_n \sim \mathcal{N}(\cdot \,|\, \mu_{c_n}, 1/\tau_{c_n}) \quad x_n \sim p(\cdot \,|\, z_n; \theta) \quad \text{for } n = 1 \dots N.$$

Here $\lambda$, $c_n$, $z_n$, and $x_n$ refer to the cluster probabilities, cluster assignments, latent codes, and observed images respectively. In the likelihood $p(x|z; \theta_x)$, we use a convolutional network to parameterize a continuous Bernoulli distribution [Loaiza-Ganem, Cunningham, 2019]. We define

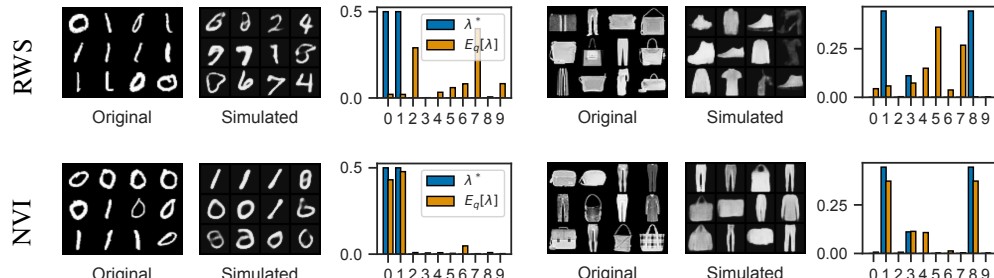

Figure 7: BGMM-VAE trained on MNIST & FashionMNIST with the RWS objective (*Top*) and the NVI objective (*Bottom*). (*Left*) Samples from a test mini-batch of size $N = 300$. (*Middle*) Samples from the generative model, generated from the $\lambda$ inferred from the test mini-batch. (*Right*) Comparison of ground truth $\lambda^*$ and the expected inferred value.

proposals $q(z|x; \phi_z)$, $q(c|z; \phi_c)$, and $q(\lambda|c; \phi_\lambda)$, which are also parameterized by neural networks. We refer to this model as Bayesian Gaussian mixture model VAE (BGMM-VAE).

**Objective.** To construct an NVI objective, we define intermediate densities for $c$ and $z$. Unlike in previous experiments, we employ tractable densities in the form of a categorical $\pi_c(c; \theta_c)$ for cluster assignments and a 8-layer planar flow $\pi_z(z; \theta_z)$ for the latent codes. Subsequently, we can define the forward and reverse densities at each step as:

$$\hat{\pi}_2(\lambda, c) = \pi(\lambda)p(c|\lambda), \quad \hat{\pi}_3(c, z) = \pi(c)p(z|c), \quad \hat{\pi}_4(z, x) = \pi(z)p(x|z),$$
$$\check{\pi}_2(\lambda, c) = \pi(c)q(\lambda|c), \quad \check{\pi}_3(c, z) = \pi(z)q(c|z), \quad \check{\pi}_4(z, x) = \tilde{\pi}(x)q(z|x),$$

where $\tilde{\pi}(x)$ is an empirical distribution over mini-batches of training data. We minimize the forward KL for the first two steps and the reverse KL at the final step.

Since the intermediate densities are tractable in this model, no nested importance sampling is required to optimize this nested objective; we can compute gradient estimates based on a (single) sample from $p(\lambda)p(c|\lambda)$ in the first term, $\pi_c(c)q(\lambda|c)$ in the second, and $\hat{p}(x)q(z|x)$ in the final term. To learn the parameters $\{\mu, \tau, \theta_x\}$ of the generative model, we maximize a single-sample approximation of a lower bound $\mathcal{L} = \mathbb{E}_q \left[ \log \left( p(x, z, c, \lambda) / q(z, c, \lambda|x) \right) \right]$.

**Results.** We evaluate NVI for the BGMM-VAE using the following procedure. We generate mini-batches with a sampled $\lambda^*$ (for which we make use of class labels that are not provided to the model). We then compute the expectation of $\lambda$ under $q(\lambda, c, z|x)$ by sampling from the inference model, and compare this value against $\lambda^*$. Additionally, we generate a new mini-batch given the inferred $\lambda$ by running the generative model forward. We compare NVI against RWS, where we use 10 samples to estimate the gradients at each iteration. The results are shown in Figure 7. The cluster indices are rearranged based on the association of clusters to true classes. We observe that for RWS, even though the sample quality is reasonable, the posterior approximation is quite poor. When we train with NVI however, both the inferred $\lambda$ and the generated samples match the test mini-batch reasonably well.

## 5 Conclusion

We developed NVI, a framework that combines nested importance sampling and variational inference by optimizing a variational objective at every level of nesting. This formulation allows us to learn proposals and intermediate densities for a general class of samplers, which admit most commonly used importance sampling strategies as special cases. Our experiments demonstrate that samplers trained with NVI are able to outperform baselines when sampling from multimodal densities, Bayesian state-space models, and hierarchical deep generative models. Moreover, our experiments show that learning intermediate distributions results in better samplers.

NVI is particularly useful in the context of deep probabilistic programming systems. Because NVI can be applied to learn proposals for a wide variety of importance samplers, it can be combined with methods for inference programming that allow users to tailor sampling strategies to a particular probabilistic program. Concretely, NVI can be applied to nested importance samplers that are defined using a grammar of composable *inference combinators* [Stites et al., 2021], functions that implement primitive operations which preserve proper weighting, such as using samples from one program as a proposal to another program, sequential composition of programs, and importance resampling.

## Acknowledgements

This work was supported by the Intel Corporation, the 3M Corporation, NSF awards 1835309 and 2047253, startup funds from Northeastern University, the Air Force Research Laboratory (AFRL), and DARPA.

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
