# A Related Work

This work draws on three lines of research. The first are importance sampling and Sequential Monte Carlo methods, many of which can be described as properly weighted nested importance samplers. The second are approaches that combine importance sampling with stochastic variational inference, either by maximizing a stochastic lower bound, or by minimizing the forward KL divergence using self-normalized estimators. Finally there are a multitude of approaches that combine stochastic variational inference with some form of forward and reverse kernel, often in the form of an MCMC transition operator. We will discuss the most directly relevant approaches in these three lines of work.

**Importance Samplers, SMC, and Proper Weighting.**    There is a vast literature on importance sampling methods, a full review of which is beyond the scope of this paper. For introductory texts on SMC methods we refer to Doucet, Johansen (2009) and Doucet et al. (2001). Much of this literature has focused on state-space models, which define a distribution over a sequence of time-dependent variables. In these models, SMC is typically used to approximate the filtering distribution at each time, which is also known as particle filtering in this context.

A limitation of SMC methods for filtering is that resampling introduces degeneracy; particles will typically coalesce to a common ancestor in $\mathcal{O}(S \log S)$ steps [Jacob et al., 2013]. This has given a rise to a literature on backwards simulation methods, which perform additional computation to reduce sample degeneracy (see Lindsten, Schön (2012) for a review). A second challenge is the estimation of global parameters of the likelihood and the transition distribution, which has given rise to a literature on particle Markov Chain Monte Carlo (PMCMC) methods [Andrieu et al., 2010; Lindsten et al., 2012; Kantas et al., 2015]. Much of this literature orthogonal to the contributions in this paper. Backward simulation and PMCMC updates preserve proper weighting, and could therefore be used in NVI, but we do not consider such approaches here.

There is also a large literature on applications of importance sampling and SMC in contexts other than state space models. SMC and its extensions have been applied to non-sequential graphical models [Naesseth et al., 2014], and are widely used to perform inference in probabilistic programs [Murray, 2013; Wood et al., 2014; Meent et al., 2015; Rainforth et al., 2016; Murray, Schön, 2018]. As we mention in the main text, two classes of methods that are directly relevant to the experiments in this paper are annealed importance sampling [Neal, 2001] and SMC samplers [Del Moral et al., 2006], which can be used to target an annealing sequence by either applying an MCMC transition operator at each step, or by defining a density on an extended space in terms of a forward and reverse kernel in the same manner as we do in this paper.

Much of this literature can be brought under a common denominator using the framework of proper weighting [Naesseth et al., 2015; Naesseth et al., 2019], which formalizes the requirements on weights that are returned by an importance sampler relative to the unnormalized density that the sampler targets. This defines an abstraction boundary, which makes it possible reason about the validity of operations at the current level of nesting in terms of the marginal of the target density at the preceding level of nesting, without having to consider the full density on the extended space for sampling operations that precede the current level. In the context of NVI, the implication of this is that we could in principle different sampling strategies at each level of nesting. We have left such approaches to future work.

**Combining Importance Sampling and Stochastic Variational Inference.**    In recent years we have seen a large number of approaches that combine stochastic variational inference with importance sampling. Much of the early work in this space was motivated by the desire to define a tighter variational lower bound. Work by Burda et al. (2016) proposed to train variational autoencoders by maximizing a stochastic lower bound $\hat{\mathcal{L}} = \log \hat{Z}$, where $\hat{Z}$ is defined by using the encoder distribution as a proposal in an importance sampler. This bound can be further tightened by using SMC rather than sequential importance sampling to compute $\hat{Z}$, which yields a lower-variance estimator $\hat{Z}$ [Le et al., 2018; Naesseth et al., 2018; Maddison et al., 2017]. More generally, tighter bounds have been proposed by using thermodynamic integration [Masrani et al., 2019] and by defining a bounds on the predictive marginal likelihood at each step [Chen et al., 2021].

Methods that make use of importance-weighted stochastic lower bounds have proven well-suited to maximum likelihood estimation, but suffer from a poor signal-to-noise ratio when performing variational inference. Rainforth et al. (2018) show that the signal-to-noise ratio for the gradient with

respect to the proposal parameters deteriorates as we increase the number of samples, which means that it is generally inadvisable to optimize a stochastic lower bound $\hat{\mathcal{L}}$ to learn these parameters. This realization has led to the development of doubly-reparameterized estimators [Tucker et al., 2018], which have since been generalized to hierarchical models and score function terms w.r.t. distribution other than the sampling distribution [Bauer, Mnih, 2021].

The realization that stochastic lower bounds are poorly suited to variational inference has also lead to a resurgence of interest in methods that derive from reweighted wake sleep [Bornschein, Bengio, 2015; Le et al., 2019], which minimize the forward KL divergence. A recent example along these lines is our own work on amortized Gibbs samplers [Wu et al., 2019], which learns proposals that approximate Gibbs kernels in an SMC sampler. This method is a special case of NVI based on the forward KL divergence at each level of nesting, in which the same target density is used at each level of nesting. Optimizing the a forward KL divergence to learn better variational proposals was also explored in recent work by Jerfel et al. (2021) using a variational boosting approach which iteratively constructs a Gaussian mixture model proposal.

Concurrent work by Arbel et al. (2021) focuses on combining AIS and SMC samplers with normalizing flows, where the reverse kernel can be chosen based on the inverse mapping defined by the flow. Similar to our work, which mainly focuses on stochastic transitions, this work optimizes a sequence of KL divergences and makes use of importance resampling between steps. However this work focuses on flow-based models and does not consider the task of learning intermediate densities.

**Combining MCMC with Stochastic Variational Inference.**   There are a large number of approaches that combine variational inference with MCMC and/or learned forward and reverse kernels, which leads to approaches that have similar use cases to the variational methods for SMC samplers that we consider in our experiments. Early work in this space by Salimans et al. (2015) computes a lower bound using Hamiltonian dynamics and defines importance weights in terms of learned forward and reverse kernels. This work ommits the Metropolis-Hastings correction step typically used with HMC due to the inability of computing the density of the transition operator. Later work by Wolf et al. (2016) incorporates an MH correction, hereby ensuring convergence to the posterior.

Work by Hoffman (2017) also initialized HMC with samples from a variational proposal but does not learn forward and reverse transition kernels; this work simply optimizes a lower bound w.r.t. the initial variational distribution and use the samples from HMC only to train the generative model. Caterini et al. (2018) combine time-inhomogeneous Hamiltonian dynamics within variational inference using an SMC sampler construction where the reverse kernel can be chosen optimally by making use of the deterministic Hamiltonian dynamics. Wang et al. (2018) develop a meta-learning approach in which samples from a ground-truth generative model serve to train variational distributions that approximate Gibbs kernels for the generative model.

Optimizing a step-wise objectives to learn forward and reverse kernels has also previously been proposed by Huang et al. (2018). However, this work differs from a SMC sampler trained with NVI in that samples are proposed from the marginal of the forward kernels as opposed to the last target density. Here, due to the intractibility of the marginal density, we can not compute and incremental importance weight to perform resampling.

## B   Notation

| | |
|---|---|
| $\pi_k(z_k)$ | $k$-th target density on $\mathcal{Z}_k$ |
| $\gamma_k(z_k) = Z_k \pi_k(z_k)$ | $k$-th unnormalized target |
| $\check{\pi}_k(z_k, z_{k-1}) = \pi_k(z_k) r_{k-1}(z_{k-1} \mid z_k, \check{\phi}_k)$ | $k$-th extended target on $\mathcal{Z}_{k-1} \times \mathcal{Z}_k$ |
| $\check{\gamma}_k(z_k, z_{k-1}) = Z_k \check{\pi}_k(z_k, z_{k-1})$ | $k$-th extended unnormalized proposal |
| $\hat{\pi}_k(z_k, z_{k-1}) = \pi_{k-1}(z_{k-1}) q_k(z_k \mid z_{k-1}, \hat{\phi}_k)$ | $k$-th extended proposal on $\mathcal{Z}_{k-1} \times \mathcal{Z}_k$ |
| $\hat{\gamma}_k(z_k, z_{k-1}) = Z_k \hat{\pi}_k(z_k, z_{k-1})$ | $k$-th extended unnormalized proposal |
| $v_k = \frac{\check{\gamma}(z_k, z_{k-1})}{\hat{\gamma}(z_k, z_{k-1})} = \frac{\gamma_k(z_k) r_{k-1}(z_{k-1} \mid z_k, \check{\phi}_k)}{\gamma_{k-1}(z_{k-1}) q_k(z_k \mid z_{k-1}, \hat{\phi}_k)}$ | $k$-th incremental weight |
| $\tilde{v}_k = v_k \frac{Z_{k-1}}{Z_k} = \frac{\pi_k(z_k) r_{k-1}(z_{k-1} \mid z_k, \check{\phi}_k)}{\pi_{k-1}(z_{k-1}) q_k(z_k \mid z_{k-1}, \hat{\phi}_k)}$ | $k$-th normalized incremental weight |
| $w_k = \prod_{k'=1}^{k} v_{k'}$ | $k$-th cumulative weight |

## C  Important Identities

**Thermodynamic Identity:**

$$\frac{d}{d\theta}\log Z_\theta = \frac{1}{Z_\theta}\frac{d}{d\theta}\int_{\mathcal{Z}_\theta} dz\,\gamma(z;\theta) = \int_{\mathcal{Z}_\theta} dz\,\frac{\gamma(z;\theta)}{Z_\theta}\frac{d}{d\theta}\log\gamma(z;\theta) = \underset{z\sim\pi(\cdot;\theta)}{\mathbb{E}}\left[\frac{\partial\log\gamma}{\partial\theta}\right]. \quad (21)$$

**Log-derivative trick a.k.a. reinforce trick:**

$$\frac{d}{d\theta}\pi(z;\theta) = \pi(z)\frac{1}{\pi(z;\theta)}\frac{d}{d\theta}\pi(z;\theta) = \pi(z)\frac{d}{d\theta}\log\pi(z;\theta) \quad (22)$$

Consequently, it holds that

$$\underset{z\sim\pi(\cdot;\theta)}{\mathbb{E}}\left[\frac{d}{d\theta}\log\pi(z;\theta)\right] = \int_{\mathcal{Z}} dz\,\pi(z;\theta)\frac{d}{d\theta}\log\pi(z;\theta) = \int_{\mathcal{Z}} dz\,\frac{d}{d\theta}\pi(z;\theta) = \frac{d}{d\theta}\int_{\mathcal{Z}} dz\,\pi(z;\theta) = 0$$

**Fisher's Identity:**

$$\nabla_\theta\log p_\theta(x) = \int dz\,p_\theta(z\mid x)\frac{d}{d\theta}\log p_\theta(x,z) \quad (23)$$

**Reweighted wake-sleep gradient for $\phi$:**

$$-\frac{\partial}{\partial\phi}\,\mathrm{KL}\left(\pi(z;\theta)\,\|\,q(z;\phi)\right) = \underset{z\sim\pi(z;\phi)}{\mathbb{E}}\left[\frac{\partial}{\partial\phi}\log q(z;\phi)\right] \quad (24)$$

## D  Connection to ELBO & EUBO

When the final the target density of interest is some posterior distribution $p(z|x)$, the NVI objective in equation 6 also defines a lower bound (or an upper bound in the case of forward KL divergence) on the $\log Z_K = \log p(x)$. Minimizing the reverse KL divergence at the last step $K$ is equivalent to maximizing the standard ELBO with variational distribution $\hat{\pi}_K$:

$$
\begin{aligned}
\mathrm{KL}\left(\hat{\pi}_K\|\check{\pi}_K\right) &= -\underset{\pi_{K-1}(z_{K-1})q_K(z_K|z_{K-1})}{\mathbb{E}}\left[\log\frac{r_{K-1}(z_{K-1}|z_K)\pi_K(z_K)}{\pi_{K-1}(z_{K-1})q_K(z_K|z_{K-1})}\right] \\
&= -\underset{\pi_{K-1}(z_{K-1})q_K(z_K|z_{K-1})}{\mathbb{E}}\left[\log\frac{r_{K-1}(z_{K-1}|z_K)p(z_K|x)}{\pi_{K-1}(z_{K-1})q_K(z_K|z_{K-1})}\right] \\
&= -\underset{\pi_{K-1}(z_{K-1})q_K(z_K|z_{K-1})}{\mathbb{E}}\left[\log\frac{r_{K-1}(z_{K-1}|z_K)p(x,z_K)}{\pi_{K-1}(z_{K-1})q_K(z_K|z_{K-1})}\right] + \log p(x),
\end{aligned}
$$

which in turn yields the following equality:

$$\log p(x) - \mathrm{KL}\left(\hat{\pi}_K\|\check{\pi}_K\right) = \underbrace{\underset{\pi_{K-1}(z_{K-1})q_K(z_K|z_{K-1})}{\mathbb{E}}\left[\log\frac{r_{K-1}(z_{K-1}|z_K)p(x,z_K)}{\pi_{K-1}(z_{K-1})q_K(z_K|z_{K-1})}\right]}_{\text{ELBO}}$$

Adding the divergences at the previous steps defines a looser lower bound on the log marginal likelihood:

$$\log p(x) - \mathrm{KL}\left(\hat{\pi}_K\|\check{\pi}_K\right) \geq \log p(x) - \mathrm{KL}\left(q_1\|\pi_1\right) - \sum_{k=2}^{K}\mathrm{KL}\left(\hat{\pi}_k\|\check{\pi}_k\right).$$

With similar derivations in the case of the forward KL divergence, we can derive an upper bound on the log marginal likelihood.

# E    Gradient estimation

To compute the gradient of the nested variational objective (NVO) we need to compute the gradients of the individual terms $D_f\left(\check{\pi}_k \,\|\, \hat{\pi}_k\right)$ w.r.t. parameters $\check{\phi}_k, \hat{\phi}_k, \theta_k$, and $\theta_{k-1}$,

$$
\frac{d\mathcal{D}}{d\hat{\phi}_k} = \frac{d D_f\left(\check{\pi}_k \,\|\, \hat{\pi}_k\right)}{d\hat{\phi}_k}, \qquad\qquad \frac{d\mathcal{D}}{d\theta_k} = \frac{d D_f\left(\check{\pi}_k \,\|\, \hat{\pi}_k\right)}{d\theta_k} + \frac{d D_f\left(\check{\pi}_{k+1} \,\|\, \hat{\pi}_{k+1}\right)}{d\theta_k},
$$

$$
\frac{d\mathcal{D}}{d\check{\phi}_k} = \frac{d D_f\left(\check{\pi}_k \,\|\, \hat{\pi}_k\right)}{d\check{\phi}_k}, \qquad\qquad \frac{d\mathcal{D}}{d\theta_{k-1}} = \frac{d D_f\left(\check{\pi}_k \,\|\, \hat{\pi}_k\right)}{d\theta_{k-1}} + \frac{d D_f\left(\check{\pi}_{k-1} \,\|\, \hat{\pi}_{k-1}\right)}{d\theta_{k-1}}.
$$

In the following we are deriving the relevant gradient terms for the general case, i.e. using an f-divergence, and state the gradient of the reverse KL-divergence, i.e. $f(w) = -\log w$, and forward KL-divergence, i.e. $f(w) = w \log w$, as special cases.

## E.1    Gradients for general f-divergences

**Gradient w.r.t. parameters $\hat{\phi}_k$ of the forward kernel.** Reparameterizing the sample $z_k \equiv z_k(\epsilon_k; \hat{\phi}_k)$ allows us, under mild conditions [3], to interchange the order of integration and differentiation and compute path-wise derivatives

$$
\frac{d}{d\hat{\phi}_k} D_f\left(\check{\pi}_k \,\|\, \hat{\pi}_k\right)
$$

$$
= \mathop{\mathbb{E}}_{z_{k-1}\sim\pi_{k-1}} \left[ \mathop{\mathbb{E}}_{\epsilon_k\sim p_k} \left[ \frac{d}{d\hat{\phi}_k} f\left( v_k \frac{Z_{k-1}}{Z_k} \right) \right] \right]
$$

$$
= \mathop{\mathbb{E}}_{z_{k-1}\sim\pi_{k-1}} \left[ \mathop{\mathbb{E}}_{\epsilon_k\sim p_k} \left[ \left.\frac{\partial f}{\partial w}\right|_{w=v_k \frac{Z_{k-1}}{Z_k}} \frac{Z_{k-1}}{Z_k} \frac{\partial v_k}{\partial z_k} \left.\frac{\partial z_k}{\partial \hat{\phi}_k}\right|_{z_k=z_k(\epsilon_k;\phi_k)} + \left.\frac{\partial f}{\partial w}\right|_{w=v_k \frac{Z_{k-1}}{Z_k}} \frac{Z_{k-1}}{Z_k} \frac{\partial v_k}{\partial \hat{\phi}_k} \right] \right]
$$

$$
= \mathop{\mathbb{E}}_{z_{k-1}\sim\pi_{k-1}} \left[ \mathop{\mathbb{E}}_{\epsilon_k\sim p_k} \left[ \left.\frac{\partial f}{\partial w}\right|_{w=v_k \frac{Z_{k-1}}{Z_k}} \frac{Z_{k-1}}{Z_k} \left( \frac{\partial v_k}{\partial z_k} \left.\frac{\partial z_k}{\partial \hat{\phi}_k}\right|_{z_k=z_k(\epsilon_k;\phi_k)} - \frac{\partial q_k}{\partial \hat{\phi}_k} \right) \right] \right]
$$

$$
= \mathop{\mathbb{E}}_{z_{k-1}\sim\pi_{k-1}} \left[ \mathop{\mathbb{E}}_{\epsilon_k\sim p_k} \left[ \left.\frac{\partial f}{\partial w}\right|_{w=v_k \frac{Z_{k-1}}{Z_k}} v_k \frac{Z_{k-1}}{Z_k} \left( \frac{\partial \log v_k}{\partial z_k} \left.\frac{\partial z_k}{\partial \hat{\phi}_k}\right|_{z_k=z_k(\epsilon_k;\phi_k)} - \frac{\partial \log q_k}{\partial \hat{\phi}_k} \right) \right] \right].
$$

Alternatively, we can compute a score function gradient which does not require the target density $\gamma_k$ to be differentiable w.r.t. the sample $z_k$ and hence can also be computed for discrete variable models.

$$
\frac{d}{d\hat{\phi}_k} D_f\left(\check{\pi}_k \,\|\, \hat{\pi}_k\right)
$$

$$
= \mathop{\mathbb{E}}_{z_{k-1}\sim\pi_{k-1}} \left[ \int_{\mathcal{Z}_k} dz_k \, \frac{d}{d\hat{\phi}_k} \left( q_k(z_k \mid z_{k-1}, \hat{\phi}_k) f\left( v_k \frac{Z_{k-1}}{Z_k} \right) \right) \right]
$$

$$
= \mathop{\mathbb{E}}_{z_{k-1}\sim\pi_{k-1}} \left[ \mathop{\mathbb{E}}_{z_k\sim q_k(\cdot|z_{k-1},\hat{\phi}_k)} \left[ f\left( v_k \frac{Z_{k-1}}{Z_k} \right) \frac{\partial \log q_k}{\partial \hat{\phi}_k} + \left.\frac{\partial f}{\partial w}\right|_{w=v_k \frac{Z_{k-1}}{Z_k}} \frac{Z_{k-1}}{Z_k} \frac{\partial v_k}{\partial \hat{\phi}_k} \right] \right]
$$

$$
= \mathop{\mathbb{E}}_{z_{k-1}\sim\pi_{k-1}} \left[ \mathop{\mathbb{E}}_{z_k\sim q_k(\cdot|z_{k-1},\hat{\phi}_k)} \left[ f\left( v_k \frac{Z_{k-1}}{Z_k} \right) \frac{\partial \log q_k}{\partial \hat{\phi}_k} + \left.\frac{\partial f}{\partial w}\right|_{w=v_k \frac{Z_{k-1}}{Z_k}} v_k \frac{Z_{k-1}}{Z_k} \frac{\partial \log v_k}{\partial \hat{\phi}_k} \right] \right]
$$

$$
= \mathop{\mathbb{E}}_{z_{k-1}\sim\pi_{k-1}} \left[ \mathop{\mathbb{E}}_{z_k\sim q_k(\cdot|z_{k-1},\hat{\phi}_k)} \left[ \left( f\left( v_k \frac{Z_{k-1}}{Z_k} \right) - \left.\frac{\partial f}{\partial w}\right|_{w=v_k \frac{Z_{k-1}}{Z_k}} v_k \frac{Z_{k-1}}{Z_k} \right) \frac{\partial \log q_k}{\partial \hat{\phi}_k} \right] \right]
$$

---

[3]These condition are given by the Leibniz integration rules

**Gradient w.r.t. parameters $\check{\phi}_k$ of the reverse kernel.** Computing the gradient w.r.t. parameters of the reverse kernel is straightforward as the expectation does not depend on parameters $\check{\phi}_k$,

$$\frac{d}{d\check{\phi}_k} \mathrm{D}_f\left(\check{\pi}_k \,\|\, \hat{\pi}_k\right)$$

$$= \mathop{\mathbb{E}}_{z_{k-1},z_k \sim \hat{\pi}_k} \left[ \frac{d}{d\check{\phi}_k} f\left(v_k \frac{Z_{k-1}}{Z_k}\right) \right]$$

$$= \mathop{\mathbb{E}}_{z_{k-1},z_k \sim \hat{\pi}_k} \left[ \left.\frac{\partial f}{\partial w}\right|_{w=v_k \frac{z_{k-1}}{Z_k}} \frac{Z_{k-1}}{Z_k} \frac{\partial v_k}{\partial \check{\phi}_k} \right]$$

$$= \mathop{\mathbb{E}}_{z_{k-1},z_k \sim \hat{\pi}_k} \left[ \left.\frac{\partial f}{\partial w}\right|_{w=v_k \frac{z_{k-1}}{Z_k}} v_k \frac{Z_{k-1}}{Z_k} \frac{\partial \log v_k}{\partial \check{\phi}_k} \right]$$

$$= \mathop{\mathbb{E}}_{z_{k-1},z_k \sim \hat{\pi}_k} \left[ \left.\frac{\partial f}{\partial w}\right|_{w=v_k \frac{z_{k-1}}{Z_k}} v_k \frac{Z_{k-1}}{Z_k} \frac{\partial \log r_{k-1}}{\partial \check{\phi}_k} \right].$$

**Gradient w.r.t. parameters $\theta_{k-1}$ of the *current proposal*.** The gradient w.r.t. $\theta_{k-1}$ requires to compute a score-function style gradient and the computation of the gradient of the log normalizing constant $\log Z_{k-1}$.

$$\frac{d}{d\theta_{k-1}} \mathrm{D}_f\left(\check{\pi}_k \,\|\, \hat{\pi}_k\right)$$

$$= \mathop{\mathbb{E}}_{z_{k-1} \sim \pi_{k-1}} \left[ \frac{\partial \log \pi_{k-1}}{\partial \theta_{k-1}} \mathop{\mathbb{E}}_{z_k \sim q_k(\cdot|z_{k-1},\hat{\phi}_k)} \left[ f\left(v_k \frac{Z_{k-1}}{Z_k}\right) \right] + \mathop{\mathbb{E}}_{z_k \sim q_k(\cdot|z_{k-1},\hat{\phi}_k)} \left[ \frac{\partial}{\partial \theta_{k-1}} f\left(v_k \frac{Z_{k-1}}{Z_k}\right) \right] \right]$$

$$= \mathop{\mathbb{E}}_{z_{k-1},z_k \sim \hat{\pi}_k} \left[ f\left(v_k \frac{Z_{k-1}}{Z_k}\right) \frac{\partial \log \pi_{k-1}}{\partial \theta_{k-1}} - \left.\frac{\partial f}{\partial w}\right|_{w=v_k \frac{z_{k-1}}{Z_k}} v_k \frac{Z_{k-1}}{Z_k} \frac{\partial \log \pi_{k-1}}{\partial \theta_{k-1}} \right]$$

$$= \mathop{\mathbb{E}}_{z_{k-1},z_k \sim \hat{\pi}_k} \left[ \left( f\left(v_k \frac{Z_{k-1}}{Z_k}\right) - \left.\frac{\partial f}{\partial w}\right|_{w=v_k \frac{z_{k-1}}{Z_k}} v_k \frac{Z_{k-1}}{Z_k} \right) \frac{\partial \log \pi_{k-1}}{\partial \theta_{k-1}} \right]$$

$$= \mathop{\mathbb{E}}_{z_{k-1},z_k \sim \hat{\pi}_k} \left[ \left( f\left(v_k \frac{Z_{k-1}}{Z_k}\right) - \left.\frac{\partial f}{\partial w}\right|_{w=v_k \frac{z_{k-1}}{Z_k}} v_k \frac{Z_{k-1}}{Z_k} \right) \left( \frac{\partial \log \gamma_{k-1}}{\partial \theta_{k-1}} - \frac{\partial \log Z_{k-1}}{\partial \theta_{k-1}} \right) \right]$$

$$= \mathop{\mathbb{E}}_{z_{k-1},z_k \sim \hat{\pi}_k} \left[ \left( f\left(v_k \frac{Z_{k-1}}{Z_k}\right) - \left.\frac{\partial f}{\partial w}\right|_{w=v_k \frac{z_{k-1}}{Z_k}} v_k \frac{Z_{k-1}}{Z_k} \right) \frac{\partial \log \gamma_{k-1}}{\partial \theta_{k-1}} \right]$$

$$\quad - \mathop{\mathbb{E}}_{z_{k-1},z_k \sim \hat{\pi}_k} \left[ \left( f\left(v_k \frac{Z_{k-1}}{Z_k}\right) - \left.\frac{\partial f}{\partial w}\right|_{w=v_k \frac{z_{k-1}}{Z_k}} v_k \frac{Z_{k-1}}{Z_k} \right) \right] \mathop{\mathbb{E}}_{z_{k-1} \sim \pi_{k-1}} \left[ \frac{\partial \log \gamma_{k-1}}{\partial \theta_{k-1}} \right]$$

$$= \mathrm{Cov}_{\hat{\pi}_k} \left[ f\left(v_k \frac{Z_{k-1}}{Z_k}\right) - \left.\frac{\partial f}{\partial w}\right|_{w=v_k \frac{z_{k-1}}{Z_k}} v_k \frac{Z_{k-1}}{Z_k}, \ \frac{\partial \log \gamma_{k-1}}{\partial \theta_{k-1}} \right]$$

$$= \mathrm{Cov}_{\hat{\pi}_k} \left[ f\left(v_k \frac{Z_{k-1}}{Z_k}\right), \ \frac{\partial \log \gamma_{k-1}}{\partial \theta_{k-1}} \right] - \mathrm{Cov}_{\hat{\pi}_k} \left[ \left.\frac{\partial f}{\partial w}\right|_{w=v_k \frac{z_{k-1}}{Z_k}} v_k \frac{Z_{k-1}}{Z_k}, \ \frac{\partial \log \gamma_{k-1}}{\partial \theta_{k-1}} \right]$$

**Gradient w.r.t. parameters $\theta_k$ of the *current target*** The gradient w.r.t. $\theta_k$ requires to compute the gradient of the log normalizing constant $\log Z_k$.

$$
\frac{d}{d\theta_k} \mathrm{D}_f \left( \check{\pi}_k \,||\, \hat{\pi}_k \right)
$$

$$
= \underset{z_{k-1}, z_k \sim \hat{\pi}_k}{\mathbb{E}} \left[ \frac{d}{d\theta_k} f \left( v_k \frac{Z_{k-1}}{Z_k} \right) \right]
$$

$$
= \underset{z_{k-1}, z_k \sim \hat{\pi}_k}{\mathbb{E}} \left[ \left. \frac{\partial f}{\partial w} \right|_{w = v_k \frac{Z_{k-1}}{Z_k}} \frac{Z_{k-1}}{Z_k} \frac{\partial v_k}{\partial \theta_k} \right]
$$

$$
= \underset{z_{k-1}, z_k \sim \hat{\pi}_k}{\mathbb{E}} \left[ \left. \frac{\partial f}{\partial w} \right|_{w = v_k \frac{Z_{k-1}}{Z_k}} v_k \frac{Z_{k-1}}{Z_k} \frac{\partial \log \pi_k}{\partial \theta_k} \right]
$$

$$
= \underset{z_{k-1}, z_k \sim \hat{\pi}_k}{\mathbb{E}} \left[ \left. \frac{\partial f}{\partial w} \right|_{w = v_k \frac{Z_{k-1}}{Z_k}} v_k \frac{Z_{k-1}}{Z_k} \left( \frac{\partial \log \gamma_k}{\partial \theta_k} - \frac{\partial \log Z_k}{\partial \theta_k} \right) \right]
$$

$$
= \underset{z_{k-1}, z_k \sim \hat{\pi}_k}{\mathbb{E}} \left[ \left. \frac{\partial f}{\partial w} \right|_{w = v_k \frac{Z_{k-1}}{Z_k}} v_k \frac{Z_{k-1}}{Z_k} \frac{\partial \log \gamma_k}{\partial \theta_k} \right] - \underset{z_{k-1}, z_k \sim \hat{\pi}_k}{\mathbb{E}} \left[ \left. \frac{\partial f}{\partial w} \right|_{w = v_k \frac{Z_{k-1}}{Z_k}} v_k \frac{Z_{k-1}}{Z_k} \right] \underset{z_k \sim \pi_k}{\mathbb{E}} \left[ \frac{\partial \log \gamma_k}{\partial \theta_k} \right]
$$

### E.2 Gradients for the reverse KL-divergence ($f(w) = -\log(w)$)

Building on the deviations for the general case derived in Appendix E.1, we derive the gradients for the reverse KL-divergence as special cases by substituting $f(w) = -\log(w)$.

**Gradient w.r.t. parameters $\hat{\phi}_k$ of the forward kernel:**
The reparameterized gradient takes the form

$$
\frac{d}{d\hat{\phi}_k} \mathrm{D}_{-\log w} \left( \check{\pi}_k \,||\, \hat{\pi}_k \right) = \underset{z_{k-1} \sim \pi_{k-1}}{\mathbb{E}} \left[ \underset{\epsilon_k \sim p_k}{\mathbb{E}} \left[ -\frac{\partial \log v_k}{\partial z_k} \frac{\partial z_k}{\partial \hat{\phi}_k} - \frac{\partial \log q_k}{\partial \hat{\phi}_k} \right] \right] \tag{25}
$$

$$
= \underset{z_{k-1} \sim \pi_{k-1}}{\mathbb{E}} \left[ \underset{\epsilon_k \sim p_k}{\mathbb{E}} \left[ -\frac{\partial \log v_k}{\partial z_k} \frac{\partial z_k}{\partial \hat{\phi}_k} \right] \right], \tag{26}
$$

whereas the score function gradient takes the form

$$
\frac{d}{d\hat{\phi}_k} \mathrm{D}_{-\log w} \left( \check{\pi}_k \,||\, \hat{\pi}_k \right) = \underset{z_{k-1} \sim \pi_{k-1}}{\mathbb{E}} \left[ \underset{z_k \sim q_k(\cdot | z_{k-1}; \hat{\phi}_k)}{\mathbb{E}} \left[ \left( 1 - \log \left( v_k \frac{Z_{k-1}}{Z_k} \right) \right) \frac{\partial \log q_k}{\partial \hat{\phi}_k} \right] \right] \tag{27}
$$

$$
= \underset{z_{k-1}, z_k \sim \hat{\pi}_k}{\mathbb{E}} \left[ -\log v_k \frac{\partial \log q_k}{\partial \hat{\phi}_k} \right]. \tag{28}
$$

The final equalities hold due to the reinforce property

$$
\underset{\epsilon_k \sim p_k}{\mathbb{E}} \left[ \left. \frac{\partial \log q_k}{\partial \hat{\phi}_k} \right|_{z_k = z_k(\epsilon, \phi)} \right] = \underset{z_k \sim q_k(\cdot | z_{k-1}, \hat{\phi}_k)}{\mathbb{E}} \left[ \frac{\partial \log q_k}{\partial \hat{\phi}_k} \right] = 0.
$$

**Gradient w.r.t. parameters $\check{\phi}$ of the reverse kernel**

$$
\frac{d}{d\check{\phi}_k} \mathrm{D}_{-\log w} \left( \check{\pi}_k \,||\, \hat{\pi}_k \right) = \underset{z_{k-1}, z_k \sim \hat{\pi}_k}{\mathbb{E}} \left[ -\frac{\partial \log r_k}{\partial \check{\phi}_k} \right]. \tag{29}
$$

**Gradient w.r.t. parameters $\theta_k$ of the *current target***

$$
\frac{d}{d\theta_k} \mathrm{D}_{-\log w} \left( \check{\pi}_k \,||\, \hat{\pi}_k \right) = \underset{z_{k-1}, z_k \sim \hat{\pi}_k}{\mathbb{E}} \left[ -\frac{\partial \log \gamma_k}{\partial \theta_k} \right] + \underset{z_k \sim \pi_k}{\mathbb{E}} \left[ \frac{\partial \log \gamma_k}{\partial \theta_k} \right]. \tag{30}
$$

**Gradient w.r.t. parameters $\theta_{k-1}$ of the *current proposal***

$$\frac{d}{d\theta_{k-1}} \mathrm{D}_{-\log w}\left(\check{\pi}_k \,\|\, \hat{\pi}_k\right) = \mathrm{Cov}_{\hat{\pi}_k}\left[-\log v_k, \frac{\partial \log \gamma_{k-1}}{\partial \theta_{k-1}}\right] \tag{31}$$

### E.3 Gradients for the forward KL-divergence ($f(w) = w\log(w)$)

First notice that

$$\mathrm{D}_{w \log w}(\check{\pi}_k \,\|\, \hat{\pi}_k) = \mathop{\mathbb{E}}_{z_{k-1}, z_k \sim \hat{\pi}_k} [w \log w] \tag{32}$$

$$= \mathop{\mathbb{E}}_{z_{k-1}, z_k \sim \check{\pi}_k} [\log w] \tag{33}$$

$$= \mathop{\mathbb{E}}_{z_{k-1}, z_k \sim \check{\pi}_k} \left[-\log w^{-1}\right] \tag{34}$$

$$= \mathrm{D}_{-\log w}(\hat{\pi}_k \,\|\, \check{\pi}_k). \tag{35}$$

Hence the gradients for the forward KL-divergence follow by symmetry from the gradient of the reverse KL-divergence by identifying the components $r_k, \pi_k$ and corresponding parameters $\hat{\phi}_k, \theta_k$ with the components of the forward density $q_k, \pi_{k-1}$ and parameters $\check{\phi}_k, \theta_{k-1}$ respectively.

**Gradient w.r.t. parameters $\hat{\phi}_k$ of the forward kernel:**

$$\frac{d}{d\hat{\phi}_k} \mathrm{D}_{w \log w}\left(\check{\pi}_k \,\|\, \hat{\pi}_k\right) = \mathop{\mathbb{E}}_{z_{k-1}, z_k \sim \check{\pi}_k}\left[-\frac{\partial \log q_k}{\partial \hat{\phi}_k}\right]$$

**Gradient w.r.t. parameters $\check{\phi}_k$ of the reverse kernel:** Note that the sample $z_{k-1}$ is assumed to be non-reparameterized. Hence we only state the score-function gradient for the forward KL-divergence.

$$\frac{d}{d\check{\phi}_k} \mathrm{D}_{w \log w}\left(\check{\pi}_k \,\|\, \hat{\pi}_k\right) = \mathop{\mathbb{E}}_{z_{k-1}, z_k \sim \check{\pi}_k}\left[\log v_k \frac{\partial \log r_k}{\partial \check{\phi}_k}\right]. \tag{36}$$

**Gradient w.r.t. parameters $\theta_k$ of the *current target***

$$\frac{d}{d\theta_k} \mathrm{D}_{w \log w}\left(\check{\pi}_k \,\|\, \hat{\pi}_k\right) = \mathrm{Cov}_{\check{\pi}_k}\left[\log v_k, \frac{\partial \log \gamma_k}{\partial \theta_k}\right] \tag{37}$$

**Gradient w.r.t. parameters $\theta_{k-1}$ of the *current proposal***

$$\frac{d}{d\theta_{k-1}} \mathrm{D}_{w \log w}\left(\check{\pi}_k \,\|\, \hat{\pi}_k\right) = \mathop{\mathbb{E}}_{z_{k-1}, z_k \sim \check{\pi}_k}\left[-\frac{\partial \log \gamma_{k-1}}{\partial \theta_{k-1}}\right] + \mathop{\mathbb{E}}_{z_k \sim \pi_{k-1}}\left[\frac{\partial \log \gamma_{k-1}}{\partial \theta_{k-1}}\right]. \tag{38}$$

Notice that the expectations of the gradients for the forward KL-divergence are w.r.t. the extended target density $\check{\pi}$ as opposed to the extended proposal $\hat{\pi}$ as it is the case for the reverse KL-divergence.

### E.4 Estimation of expectations w.r.t. intermediate target densities.

When estimating an expectation of some function $h$ w.r.t. an intermediate extended proposal $\hat{\pi}_k$, we can rewrite the expectation w.r.t. properly weighted samples from the previous level of nesting using Definition 2.1

$$\mathop{\mathbb{E}}_{z_{k-1}, z_k \sim \hat{\pi}_k} [h(z_{k-1}, z_k)] = \mathop{\mathbb{E}}_{w_{k-1}, z_{k-1} \sim \Pi_{k-1}}\left[\frac{w_{k-1}}{cZ_{k-1}} \mathop{\mathbb{E}}_{z_k \sim q_k(\cdot|z_{k-1}, \hat{\phi}_k)}\left[h(z_{k-1}, z_k)\right]\right]. \tag{39}$$

Here, $\Pi_k$ denotes the probability density over weighted samples $(z_k, w_k)$ of the nested importance sampler from the $k$-th level of nesting. In our experiments we have $c = 1$. Similarly, we can rewrite

expectation w.r.t. intermediate extended target density $\check{\pi}_k$

$$\mathop{\mathbb{E}}_{z_{k-1}, z_k \sim \check{\pi}_k} [h(z_{k-1}, z_k)] = \mathop{\mathbb{E}}_{z_{k-1}, z_k \sim \hat{\pi}_k} \left[ \frac{Z_{k-1}}{Z_k} v_k \, h(z_{k-1}, z_k) \right] \tag{40}$$

$$= \mathop{\mathbb{E}}_{w_{k-1}, z_{k-1} \sim \Pi_{k-1}} \left[ \frac{w_{k-1}}{cZ_{k-1}} \mathop{\mathbb{E}}_{z_k \sim q_k(\cdot | z_{k-1}, \hat{\phi}_k)} \left[ \frac{Z_{k-1}}{Z_k} v_k \, h(z_{k-1}, z_k) \right] \right] \tag{41}$$

$$= \mathop{\mathbb{E}}_{w_{k-1}, z_{k-1} \sim \Pi_{k-1}} \left[ \frac{w_{k-1} v_k}{cZ_k} \mathop{\mathbb{E}}_{z_k \sim q_k(\cdot | z_{k-1}, \hat{\phi}_k)} \left[ h(z_{k-1}, z_k) \right] \right] . \tag{42}$$

The resulting expressions can be approximated using a self-normalized estimator as stated in Equation 2. In NVI, we assume that samples $(z_k, w_k) \sim \Pi_k$ are non-reparameterized and hence do not *carry back* gradient to the previous level of nesting. In practice, these samples might be generated using a reparameterized forward kernel and hence their gradient has to be detached.

## F  NVI Algorithm Block

---

**Algorithm 1:** Nested Variational Inference

---

**Input:** $\{\pi_k(z_k; \theta_k)\}_{k=1}^K, q_1(z_1, \hat{\phi}_1), \{q_{k+1}(z_{k+1} | z_k; \hat{\phi}_{k+1}), r_k(z_k | z_{k+1}; \check{\phi}_k)\}_{k=1}^{K-1}$
$\rho = \{\theta_k\}_{k=1}^K \cup \{\hat{\phi}_{k+1}, \check{\phi}_{k-1}\}_{k=2}^K$
**while** *not converged* **do**
    $z_1 \sim q_1(\cdot; \hat{\phi}_1)$
    $\log w_1 \leftarrow \log \frac{\gamma_1(z_1)}{q_1(z_1)}$
    $\Delta\rho \leftarrow \nabla_{\rho_1} D_f(\pi_q \,\|\, q_1)$
    **for** $k = 2 \cdots K$ **do**
        **if** *resample* **then**
            $z_{k-1}, w_{k-1} \leftarrow \text{resample}(w_{k-1})$
        $z_{k-1} \leftarrow \perp z_{k-1}$ ;                      `// detach previous sample`
        $z_k \sim q_k(\cdot | z_{k-1}; \hat{\phi}_k)$
        $\log v_k \leftarrow \log \frac{\hat{\gamma}_k(z_{k-1}, z_k)}{\check{\gamma}_k(z_{k-1}, z_k)}$
        $\log w_k \leftarrow \log w_{k-1} + \log v_k$
        $\Delta\rho \leftarrow \Delta\rho + \nabla_{\hat{\rho}_k} D_f(\check{\pi}_k \,\|\, \hat{\pi}_k)$ ;      `// based on Eq. 9 or Eq. 12`
        $\Delta\rho \leftarrow \Delta\rho + \nabla_{\check{\rho}_k} D_f(\check{\pi}_k \,\|\, \hat{\pi}_k)$ ;      `// based on Eq. 10 or Eq. 13`
    **end**
    $\rho \leftarrow \rho + \eta \Delta\rho$
**end**

---

## G  Experiment Details

### G.1  Experiment 1: Annealing

We are targeting an unnormalized Gaussian mixture model (GMM)

$$\gamma_K(z_K) = \sum_{m=1}^{M} \mathcal{N}(z_K; \ \mu_m, \ \sigma^2 I_{2\times 2}), \qquad \mu_m = \left( r\sin\left(\frac{2m\pi}{M}\right), r\cos\left(\frac{2m\pi}{M}\right)\right),$$

with $M = 8$ equidistantly spaced modes with variance $\sigma^2 = 0.5$ along a circle with radius $r = 10$.

We do not perform an initial importance sampling step in this experiment and define the initial proposal density $q_1 := \pi_1$ to be a multivariate normal with mean $0$ and standard deviation $5$. We model the transition kernels to be conditional normal,

$$q_k(z_k \mid z_{k-1}) = \mathcal{N}(z_k; \ \mu_k(z_{k-1}), \sigma_k(z_{k-1})^2 \mathcal{I}_{2\times 2}),$$
$$r_{k-1}(z_{k-1} \mid z_k) = \mathcal{N}(z_{k-1}; \ \mu_k(z_k), \sigma_k(z_k)^2 \mathcal{I}_{2\times 2}).$$

with mappings for the mean $\mu$ and standard deviation $\sigma$ as follows:

$$\mu(z) = W_\mu^T \left(h(z) + z\right) + b_\mu, \quad \sigma(z) = \mathrm{softplus}(W_\sigma^T h(z) + b_\sigma) \quad h(z) = W_h^T z + b_h. \quad (43)$$

In the experiments the hidden layer consists of $50$ neurons, i.e. $h(z) \in \mathbb{R}^{50}$. For the flow-based models the kernels are specified by the flow and hence fully deterministic. In this case the incremental importance weight simplifies to

$$v_k = \frac{\gamma_{k+1}(z_{k+1})}{q_k(z_{k+1})} = \frac{\gamma_{k+1}(z_{k+1})}{\gamma_k(z_k)\log|J_{f_k^{-1}}(z_{k+1})|} = \frac{\gamma_{k+1}(f_k(z_k))}{\gamma_k(z_k)\log|J_{f_k}(z_k)|^{-1}}, \quad (44)$$

where the mapping $f_k$ is a planar flow consisting of $32$ layers. All methods are trained for $20,000$ iteration using the Adam optimizer with a learning rate of $1\mathrm{e}^{-3}$. In AVO and NVI methods we observed very small standard deviations across runs for the log normalizing constant and the ESS, which is why we did not include them in Table 1. This excludes flow-based methods for $K = 2$ (one-step flow) where we observed significant variance in standard between runs (AVO-flow: $28 \pm 12$, NVIR*-flow: $28 \pm 11$).

Figure 8 shows examplary samples from flow based models trained with AVO-flow and NVIR*-flow.

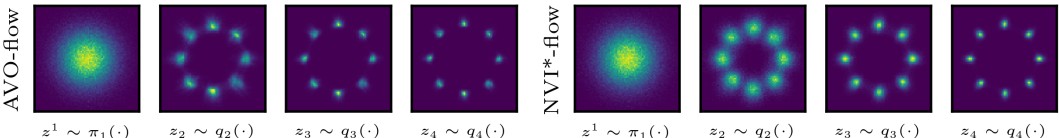

Figure 8: Samples from forward kernels trained with AVO, and NVIR*.

### G.2  Experiment 2: Learning Heuristic Factors for state-space models

We evaluate NVI for a hidden Markov model (HMM) where the likelihood is given by a mixture of Gaussians (GMM),

$$\tau_m, \mu_m \sim p(\tau, \mu) = \mathrm{NormalGamma}(\alpha_0, \beta_0, \mu_0, \nu_0), \qquad m = 1, 2, ..., M,$$
$$z_1 \sim p(z_1) = \mathrm{Categorical}(\pi),$$
$$z_k \mid z_{k-1} = m \sim p(z_k \mid z_{k-1}) = \mathrm{Categorical}(A_m),$$
$$x_k \mid z_k = m \sim p(x_k \mid z_k, \tau, \mu) = \mathrm{Normal}(\mu_m, \sigma_m), \qquad k = 1, 2, ..., K.$$

where $M$ is the number of clusters and $K$ is the number of time steps in a HMM instance; $z_{1:K}$ and $x_{1:K}$ are the discrete hidden states and observations respectively, $\eta := \{\tau_m, \mu_m\}_{m=1}^{M}$ is the set of global variables, where $\tau_m$, and $\mu_m$ are precision and mean of the m-th GMM cluster respectively. We choose $M = 4$, $K = 200$, and the hyperparameters as follows

$$\alpha_0 = 8.0, \qquad \beta = 8.0, \qquad \mu_0 = 0.0, \qquad \nu_0 = 0.001, \qquad \pi = (0.25, 0.25, 0.25, 0.25). \quad (45)$$

$$A = \begin{bmatrix} \frac{9}{10} & \frac{1}{30} & \frac{1}{30} & \frac{1}{30} \\ \frac{1}{30} & \frac{9}{10} & \frac{1}{30} & \frac{1}{30} \\ \frac{1}{30} & \frac{1}{30} & \frac{9}{10} & \frac{1}{30} \\ \frac{1}{30} & \frac{1}{30} & \frac{1}{30} & \frac{9}{10} \end{bmatrix}$$

To perform inference, we learn the following proposals in form of

$$q_\phi(\mu_{1:M}, \tau_{1:M} \mid x_{1:K}) = \prod_{m=1}^M \text{NormalGamma}(\tilde{\alpha}_m, \tilde{\beta}_m, \tilde{\mu}_m, \tilde{\nu}_m), \tag{46}$$

$$q_\phi(z_1 \mid x_1, \mu_{1:M}, \tau_{1:M}) = \text{Categorical}(\tilde{\pi}_1), \tag{47}$$

$$q_\phi(z_k \mid x_k, z_{k-1}, \tau_{1:M}, \mu_{1:M}) = \text{Categorical}(\tilde{\pi}_k). \tag{48}$$

We use the tilde symbol ($\tilde{\ }$) to denote the parameters of the variational distributions that are the outputs of the neural networks. We also use NVI to learn a heuristic factor $\psi_\theta(x_{k:K} \mid \tau, \mu)$ in form of

$$\psi_\theta(x_{k:K} \mid \tau, \mu) = \prod_{l=k}^K \sum_{m=1}^M \text{Normal}(x_l; \tau_m, \mu_m)\psi_\theta^{\text{MLP}}(z_l = m \mid \eta, x_l).$$

In addition we consider a baseline that uses a Gaussian mixture model as hand-coded heuristic in form of

$$\psi_{\text{GMM}}(x_{k:K} \mid \tau, \mu) = \prod_{l=k}^K \sum_{m=1}^M \text{Normal}(x_l; \tau_m, \mu_m)p(z_l = m),$$

We optimize the parameters $\{\phi, \theta\}$ by minimizing forward KL divergences defines as follows

$$\begin{aligned}
\mathcal{L}_0(\theta, \phi) &= \text{KL}\left(\check{\pi}_{0,\theta}(\eta) \,\|\, q_\phi(\eta | x_{1:T})\right) \\
&= \text{KL}\left(\pi_{0,\theta}(\eta) \,\|\, q_\phi(\eta | x_{1:T})\right), \\
\mathcal{L}_1(\theta, \phi) &= \text{KL}\left(\check{\pi}_{1,\theta}(z_1, \eta) \,\|\, \hat{\pi}_{1,\theta}(z_1, \eta)\right) \\
&= \text{KL}\left(\pi_{1,\theta}(z_1, \eta) \,\|\, \pi_{0,\theta}(\eta)q_\phi(z_1 | x_1, \eta)\right), \\
\mathcal{L}_k(\theta, \phi) &= \text{KL}\left(\check{\pi}_{k,\theta}(z_{1:k}, \eta) \,\|\, \hat{\pi}_{k,\theta}(z_{1:k}, \eta)\right) \\
&= \text{KL}\left(\pi_{k,\theta}(z_{1:k}, \eta) \,\|\, \pi_{k-1,\theta}(z_{1:k-1}, \eta)q_\phi(z_t | x_t, z_{k-1}, \eta)\right), \qquad k = 2, 3, ..., K.
\end{aligned}$$

In NVI, we learn heuristic factors $\psi_\theta$ that approximate the marginal likelihood of future observations. We define a sequence of densities $(\gamma_0, \ldots, \gamma_K)$,

$$\gamma_0(\eta) = p(\eta)\,\psi_\theta(x_{1:K}|\eta), \quad \gamma_k(z_{1:k}, \eta) = p(x_{1:k}, z_{1:k}, \eta)\,\psi_\theta(x_{k+1:K} \mid \eta), \quad k = 1, 2, ..., K.$$

In practice we found that partial optimization (i.e. only taking gradient w.r.t the right hand side of each KL) yields better performance compared to the full optimization of the objective. We will derive the gradient for each case.

### G.2.1 Partial optimization.

We consider only taking gradient w.r.t. the right hand side of each KL divergence.

When $k = 0$, we have

$$-\nabla_\phi \mathcal{L}_0(\phi) = -\nabla_\phi \text{KL}\left(\pi_0(\eta) \,\|\, q_\phi(\eta \mid x_{1:T})\right) = \mathbb{E}_{\pi_0}\left[\nabla \log q_\phi(\eta \mid x_{1:T})\right]. \tag{49}$$

When $k = 1 : K$, we have

$$\begin{aligned}
-\nabla_{\phi,\theta}\mathcal{L}_k(\phi, \theta) &= -\nabla_{\phi,\theta}\text{KL}\left(\pi_k(z_{1:k}, \eta) \,\|\, \pi_{k-1,\theta}(z_{1:k-1}, \eta)q_\phi(z_t \mid x_k, z_{k-1}, \eta)\right) \tag{50} \\
&= \mathbb{E}_{\pi_k}\left[\nabla \log \pi_{k-1,\theta}(z_{1:k-1}, \eta) + \nabla \log q_\phi(z_t \mid x_k, z_{k-1}, \eta)\right] \tag{51} \\
&= \mathbb{E}_{\pi_k}\left[\nabla \log \psi_\theta(x_{k:K} \mid \eta) + \nabla \log q_\phi(z_t \mid x_k, z_{k-1}, \eta)\right] \tag{52} \\
&\quad - \mathbb{E}_{\pi_{k-1}}\left[\nabla \log \psi_\theta(x_{k:K} \mid \eta)\right] \tag{53}
\end{aligned}$$

**Full optimization.** Now we consider taking gradient w.r.t. the full objective.

When $k = 0$,

$$-\nabla_{\phi,\theta}\mathcal{L}_0(\phi,\theta) = -\nabla_{\phi,\theta}\mathrm{KL}\left(\pi_{0,\theta}(\eta) \,||\, q_\phi(\eta \mid x_{1:K})\right) \tag{54}$$

$$= \mathbb{E}_{\pi_0}\left[-\nabla \log \pi_{0,\theta}(\eta \mid x_{1:K}) + \nabla \log q_\phi(\eta \mid x_{1:K})\right] \tag{55}$$

$$+ \mathbb{E}_{\pi_0}\left[-\nabla \log \pi_{0,\theta}(\eta \mid x_{1:T})\left(\log \frac{p(\eta)\psi_\theta(x_{1:K} \mid \eta)}{q_\phi(\eta \mid x_{1:K})} - \log Z_0\right)\right], \tag{56}$$

$$= \mathbb{E}_{\pi_0}\left[\nabla \log q_\phi(\eta \mid x_{1:K})\right] - \mathbb{E}_{\pi_0}\left[\log \frac{p(\eta)\psi_\theta(x_{1:K} \mid \eta)}{q_\phi(\eta \mid x_{1:K})}\nabla \log \psi_\theta(x_{1:K} \mid \eta)\right] \tag{57}$$

$$+ \mathbb{E}_{\pi_0}\left[\log \frac{p(\eta)\psi_\theta(x_{1:K} \mid \eta)}{q_\phi(\eta \mid x_{1:K})}\right]\mathbb{E}_{\pi_0}\left[\nabla \log \psi_\theta(x_{1:K} \mid \eta)\right] \tag{58}$$

If $k = 1 : K$,

$$-\nabla_{\phi,\theta}\mathcal{L}_k(\phi,\theta) \tag{59}$$

$$= -\nabla_{\phi,\theta}\mathrm{KL}\left(\pi_{k,\theta}(z_{1:k},\eta) \,||\, \pi_{k-1,\theta}(z_{1:k-1},\eta)q_\phi(z_k \mid x_k, z_{k-1},\eta)\right) \tag{60}$$

$$= \mathbb{E}_{\pi_k}\left[-\nabla \log \frac{\pi_{k,\theta}(z_{1:k},\eta)}{\pi_{k-1,\theta}(z_{1:k-1},\eta)\,q_\phi(z_k \mid x_k, z_{k-1},\eta)}\right] \tag{61}$$

$$+ \mathbb{E}_{\pi_k}\left[\log \frac{\pi_{k,\theta}(z_{1:k},\eta)}{\pi_{k-1,\theta}(z_{1:k-1},\eta)\,q_\phi(z_k \mid x_k, z_{k-1},\eta)}\left(-\nabla \log \pi_{k,\theta}(z_{1:k},\eta)\right)\right] \tag{62}$$

$$= \mathbb{E}_{\pi_k}\left[\nabla \log \psi_\theta(x_{k:K} \mid \eta) + \nabla \log q_\phi(z_k \mid x_k, z_{k-1},\eta)\right] - \nabla \log Z_{k-1} \tag{63}$$

$$+ \mathbb{E}_{\pi_k}\left[\log \frac{\psi_\theta(x_{k+1:K} \mid \eta)}{\psi_\theta(x_{k:K} \mid \eta)\,q_\phi(z_k \mid x_k, z_{k-1},\eta)}\left(-\nabla \log \pi_k(z_{1:k},\eta)\right)\right], \tag{64}$$

$$= \mathbb{E}_{\pi_k}\left[\nabla \log \psi_\theta(x_{k:K} \mid \eta) + \nabla \log q_\phi(z_k \mid x_k, z_{k-1},\eta)\right] - \mathbb{E}_{\pi_{k-1}}\left[\nabla \log \psi_\theta(x_{k:K} \mid \eta)\right] \tag{65}$$

$$+ \mathbb{E}_{\pi_k}\left[\log \frac{\psi_\theta(x_{k+1:K} \mid \eta)}{\psi_\theta(x_{k:K} \mid \eta)\,q_\phi(z_k \mid x_k, z_{k-1},\eta)}\left(-\nabla \log \psi_\theta(x_{k+1:K} \mid \eta)\right)\right] \tag{66}$$

$$+ \mathbb{E}_{\pi_k}\left[\nabla \log \psi_\theta(x_{k+1:K} \mid \eta)\right]\mathbb{E}_{\pi_k}\left[\log \frac{\psi_\theta(x_{k+1:K} \mid \eta)}{\psi_\theta(x_{k:K} \mid \eta)\,q_\phi(z_k \mid x_k, z_{k-1},\eta)}\right] \tag{67}$$

### G.2.2 Architectures of the Proposals and Heuristic Factor

For the neural proposals, we employ the neural parameterizations based on the neural sufficient statistics [Wu et al., 2019]. We will discuss each of them in the following.

**Proposal for the global variables.**

$$q_\phi(\mu_{1:M}, \tau_{1:M} \mid x_{1:K}) = \prod_{m=1}^M \mathrm{NormalGamma}(\tilde{\alpha}_m, \tilde{\beta}_m, \tilde{\mu}_m, \tilde{\nu}_m)$$

We firstly feed each $x_k$ into a MLP to predict pointwise features, also known as neural sufficient statistics [Wu et al., 2019]

| Input $x_k \in \mathbb{R}^1$ |
| --- |
| FC 128. Tanh. |
| FC 4. Softmax. $(t_{k,1}, t_{k,2}, t_{k,3}, t_{k,4})$ |

Then we aggregate over all points and compute the intermediate-level features for each of the clusters,

$$H_m = \left(\sum_k t_{k,m}, \sum_k t_{k,m}\, x_k, \sum_k t_{k,m}\, x_k^2\right), \qquad m = 1, 2, 3, 4.$$

We feed these features into a MLP to predict the parameters of the variational distribution,

| Input $H_m \in \mathbb{R}^3$ | | | |
| --- | --- | --- | --- |
| FC 128. Tanh. | FC 128. Tanh. | FC 128. Tanh. | FC 128. Tanh. |
| FC 1. Exp(). $(\tilde{\alpha}_m)$ | FC 1. Exp(). $(\tilde{\beta}_m)$ | FC 1. $(\tilde{\mu}_m)$ | FC 1. Exp(). $(\tilde{\nu}_m)$ |

where Exp() means that we exponentiate the corresponding output values.

**Proposal for the initial state.**

$$q_\phi(z_1 \mid x_1, \tau_{1:M}, \mu_{1:M}) = \text{Categorical}(\tilde{\pi}_1)$$

We concatenate each $x_k$ with each of the cluster parameters and then predict logits as the assignments, followed by a softmax normalization. Then we normalize the logits as

| Input $x_k \in \mathbb{R}^1$, $\mu_m \in \mathbb{R}^1$, $\tau_m \in \mathbb{R}^1_+$ |
| --- |
| Concatenate$[x_k \ \mu_m, \ \tau_m]$ |
| FC 128. Tanh. FC 1. $(\tilde{\pi}_{k,m})$ |

$$\tilde{\pi}_k = \text{Softmax}\left( \tilde{\pi}_{k,1}, \tilde{\pi}_{k,2}, \tilde{\pi}_{k,3}, \tilde{\pi}_{k,4} \right).$$

**Proposal for the forward transitional state.**

$$q_\phi(z_k \mid x_k, z_{k-1}, \tau_{1:M}, \mu_{1:M}) = \text{Categorical}(\tilde{\pi}_k)$$

This is similar to the initial proposal, except that we concatenate the previous state as the input. We

| Input $x_k \in \mathbb{R}^1$, $z_{k-1} \in \mathbb{R}^1$, $\mu_m \in \mathbb{R}^1$, $\tau_m \in \mathbb{R}^1_+$ |
| --- |
| Concatenate$[x_k \ z_{k-1}, \ \mu_m, \ \tau_m]$ |
| FC 128. Tanh. FC 1. ($\tilde{\pi}_{k,m}$) |

then normalize the logits using a softmax,

$$\tilde{\pi}_k = \text{Softmax}\left(\tilde{\pi}_{k,1}, \tilde{\pi}_{k,2}, \tilde{\pi}_{k,3}, \tilde{\pi}_{k,4}\right).$$

**Heuristic factor $\psi_\theta(x_{k:K} \mid \eta)$.**
The neural heuristic factor takes as input the concatenation of each point and each of the cluster parameters, and output the logits

| Input $x_k \in \mathbb{R}^1$, $\mu_m \in \mathbb{R}^1$, $\tau_m \in \mathbb{R}^1_+$ |
| --- |
| Concatenate$[x_k \ \mu_m, \ \tau_m]$ |
| FC 128. Tanh. FC 1. ($\psi_\theta(z = m \mid x_k, \tau_m, \mu_k)$) |

### G.3 Experiment 3: BGMM-VAE

We train on the datasets MNIST and FashionMNIST in which we sample mini-batches of size $N = 10$ such that the classes are distributed based on a Dirichlet distribution with the same $\alpha$ as the generative model (We set $\alpha = 0.5$ in our experiments). In NVI, we used a higher $N$ for the first two KLs (this is only applicable to NVI because the KLs are optimized locally). For RWS, we used 10 samples per $x_n$. We trained all models for 50k iterations, and we used 20 mini-batches per iteration to estimate the overall objective. Fro the optimizers, we used Adam with learning rate 1e-3 for the $q(x|z, \theta_x)$ and $r(z|x, \phi_z)$ and 5e-2 for all models.

**Optimization**  In this experiment, we only optimize the intermediate only at the target level at each level of nesting. For example, in the 1st step, we optimize

$$\min \text{KL}\left(\hat{\pi}_1(\lambda, c) \| \check{\pi}_1(\lambda, c)\right) := \min \text{KL}\left(p(\lambda)p(c|\lambda) \| \pi(c), q(\lambda|c)\right)$$

with respect to $\pi(c)$. This way $\pi(c)$ is optimized to match the marginal distribution $p(c)$ of the generative model. In the next step, we detach the parameters for $p(c)$ and only optimize to $q$ and $\pi(z)$. Assuming $\pi(c) \approx p(c)$, optimizing the 2nd step in this manner leads $\pi(z)$ to match the marginal distribution $p(z)$ of the generative model.

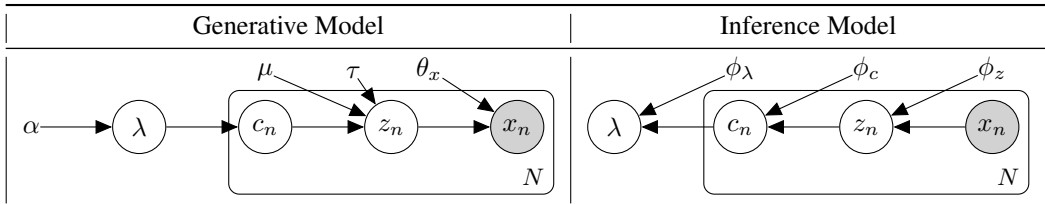

Figure 9: Overview of BGMM-VAE.

## References

Andrieu, Christophe, Doucet, Arnaud, Holenstein, Roman. "Particle Markov Chain Monte Carlo Methods". *Journal of the Royal Statistical Society: Series B (Statistical Methodology)* 72.3 (2010), pp. 269–342.

| Encoder $q(z|x; \phi_z)$ | Decoder $p(x|z, \theta_x)$ |
|---|---|
| Input $x \in \mathbb{R}^{1 \times 28 \times 28}$ | Input $z \in \mathbb{R}^{10}$ |
| Conv 32. $4 \times 4$, Stride 2, SiLU activation
Conv 64. $4 \times 4$, Stride 2, SiLU activation
Conv 64. $4 \times 4$, Stride 2, SiLU activation
Conv 64. $4 \times 4$, Stride 2, SiLU activation
FC $2 \times 10$ | FC 256. SiLU activation
UpConv 64. $4 \times 4$, Stride 2, SiLU activation
UpConv 64. $4 \times 4$, Stride 2, SiLU activation
UpConv 64. $4 \times 4$, Stride 2, SiLU activation
UpConv 32. $4 \times 4$, Stride 2, Sigmoid activation |

| $q(c|z, \phi_c)$ | $q(\lambda|c, \phi_\lambda)$ |
|---|---|
| Input $z \in \mathbb{R}^{10}$ | Input $\{c_n\}_{n=1}^N \quad c_n \in \mathbb{N}^+$ |
| FC 64. SiLU activation
FC 64. SiLU activation
FC 10 (# clusters)
Categorical(output) | FC 64. SiLU activation
Mean() (over $N$)
FC 64. SiLU activation
FC 10. SoftPlus activation (# clusters)
Dirichlet(output + $\sum_c$ c_onehot) |

Figure 10: BGMM-VAE architectures.

Arbel, Michael, Matthews, Alexander GDG, Doucet, Arnaud. "Annealed Flow Transport Monte Carlo". *arXiv preprint arXiv:2102.07501* (2021).

Bauer, Matthias, Mnih, Andriy. "Generalized Doubly Reparameterized Gradient Estimators". *arXiv preprint arXiv:2101.11046* (2021).

Bornschein, Jörg, Bengio, Yoshua. "Reweighted Wake-Sleep". *International Conference on Learning Representations* (2015). arXiv: 1406.2751.

Burda, Yuri, Grosse, Roger, Salakhutdinov, Ruslan. "Importance Weighted Autoencoders". *International Conference on Representations*. 2016. arXiv: 1509.00519.

Caterini, Anthony L, Doucet, Arnaud, Sejdinovic, Dino. "Hamiltonian variational auto-encoder". *arXiv preprint arXiv:1805.11328* (2018).

Chen, Shuangshuang, Ding, Sihao, Karayiannidis, Yiannis, Bjorkman, Marten. "Monte Carlo Filtering Objectives: A New Family of Variational Objectives to Learn Generative Model and Neural Adaptive Proposal for Time Series". *arXiv preprint arXiv:2105.09801* (2021).

Del Moral, Pierre, Doucet, Arnaud, Jasra, Ajay. "Sequential Monte Carlo Samplers". *Journal of the Royal Statistical Society: Series B (Statistical Methodology)* 68.3 (June 2006), pp. 411–436. DOI: 10.1111/j.1467-9868.2006.00553.x.

Doucet, Arnaud, Freitas, Nando, Gordon, Neil, eds. *Sequential Monte Carlo Methods in Practice*. New York, NY: Springer New York, 2001. ISBN: 978-1-4419-2887-0 978-1-4757-3437-9. DOI: 10.1007/978-1-4757-3437-9.

Doucet, Arnaud, Johansen, Adam M. "A tutorial on particle filtering and smoothing: Fifteen years later". *Handbook of nonlinear filtering* 12.656-704 (2009), p. 3.

Hoffman, Matthew D. "Learning deep latent Gaussian models with Markov chain Monte Carlo". *Proceedings of the 34th International Conference on Machine Learning-Volume 70*. JMLR. org. 2017, pp. 1510–1519.

Huang, Chin-Wei, Tan, Shawn, Lacoste, Alexandre, Courville, Aaron C. "Improving explorability in variational inference with annealed variational objectives". *Advances in Neural Information Processing Systems*. 2018, pp. 9701–9711.

Jacob, Pierre E., Murray, Lawrence M., Rubenthaler, Sylvain. "Path Storage in the Particle Filter". *Statistics and Computing* (Dec. 2013). DOI: 10.1007/s11222-013-9445-x.

Jerfel, Ghassen, Wang, Serena Lutong, Fannjiang, Clara, Heller, Katherine A, Ma, Yian, Jordan, Michael. "Variational Refinement for Importance SamplingUsing the Forward Kullback-Leibler Divergence". *Third Symposium on Advances in Approximate Bayesian Inference*. 2021.

Kantas, Nikolas, Doucet, Arnaud, Singh, Sumeetpal S., Maciejowski, Jan, Chopin, Nicolas. "On Particle Methods for Parameter Estimation in State-Space Models". *Statistical Science* 30.3 (Aug. 2015), pp. 328–351. ISSN: 0883-4237, 2168-8745. DOI: 10.1214/14-STS511.

Le, Tuan Anh, Igl, Maximilian, Rainforth, Tom, Jin, Tom, Wood, Frank. "Auto-Encoding Sequential Monte Carlo". *International Conference on Learning Representations*. 2018. arXiv: 1705.10306.

Le, Tuan Anh, Kosiorek, A, Siddharth, N, Teh, Yee Whye, Wood, Frank. "Revisiting reweighted wake-sleep for models with stochastic control flow" (2019).

Lindsten, Fredrik, Jordan, Michael I, Schön, Thomas B. "Ancestor Sampling for Particle Gibbs". *Advances in Neural Information Processing Systems*. Oct. 2012, pp. 2591–2599.

Lindsten, Fredrik, Schön, Thomas B. "On the Use of Backward Simulation in the Particle Gibbs Sampler". *2012 IEEE International Conference on Acoustics, Speech and Signal Processing (ICASSP)*. Vol. 1. IEEE, Mar. 2012, pp. 3845–3848. ISBN: 978-1-4673-0046-9. DOI: 10.1109/ICASSP.2012.6288756.

Maddison, Chris J, Lawson, Dieterich, Tucker, George, Heess, Nicolas, Norouzi, Mohammad, Mnih, Andriy, Doucet, Arnaud, Teh, Yee Whye. "Filtering variational objectives". *Proceedings of the 31st International Conference on Neural Information Processing Systems*. 2017, pp. 6576–6586.

Masrani, Vaden, Le, Tuan Anh, Wood, Frank. "The Thermodynamic Variational Objective". *Advances in Neural Information Processing Systems*. Ed. by H. Wallach, H. Larochelle, A. Beygelzimer, F. d'Alché-Buc, E. Fox, R. Garnett. Vol. 32. Curran Associates, Inc., 2019. URL: https://proceedings.neurips.cc/paper/2019/file/618faa1728eb2ef6e3733645273ab145-Paper.pdf.

Meent, Jan-Willem van de, Yang, Hongseok, Mansinghka, Vikash, Wood, Frank. "Particle Gibbs with Ancestor Sampling for Probabilistic Programs". *Artificial Intelligence and Statistics*. 2015. arXiv: 1501.06769.

Murray, Lawrence M. "Bayesian State-Space Modelling on High-Performance Hardware Using LibBi". *arXiv:1306.3277 [stat]* (June 2013). arXiv: 1306.3277 [stat].

Murray, Lawrence M., Schön, Thomas B. "Automated Learning with a Probabilistic Programming Language: Birch". *Annual Reviews in Control* 46 (Jan. 2018), pp. 29–43. ISSN: 1367-5788. DOI: 10/ghxhx2.

Naesseth, Christian, Linderman, Scott, Ranganath, Rajesh, Blei, David. "Variational sequential monte carlo". *International Conference on Artificial Intelligence and Statistics*. PMLR. 2018, pp. 968–977.

Naesseth, Christian, Lindsten, Fredrik, Schon, Thomas. "Nested Sequential Monte Carlo Methods". *International Conference on Machine Learning*. 2015, pp. 1292–1301.

Naesseth, Christian A., Lindsten, Fredrik, Schön, Thomas B. "Sequential Monte Carlo for Graphical Models". *arXiv:1402.0330 [stat]* (Oct. 2014). arXiv: 1402.0330 [stat].

Naesseth, Christian A., Lindsten, Fredrik, Schön, Thomas B. "Elements of Sequential Monte Carlo". *arXiv:1903.04797 [cs, stat]* (Mar. 2019). arXiv: 1903.04797. (Visited on 12/16/2019).

Neal, Radford M. "Annealed importance sampling". *Statistics and computing* 11.2 (2001), pp. 125–139.

Rainforth, Tom, Kosiorek, Adam, Le, Tuan Anh, Maddison, Chris, Igl, Maximilian, Wood, Frank, Teh, Yee Whye. "Tighter Variational Bounds Are Not Necessarily Better". en. *International Conference on Machine Learning*. July 2018, pp. 4277–4285.

Rainforth, Tom, Naesseth, Christian A., Lindsten, Fredrik, Paige, Brooks, Meent, Jan-Willem van de, Doucet, Arnaud, Wood, Frank. "Interacting Particle Markov Chain Monte Carlo". *Proceedings of The 33rd International Conference on Machine Learning,* 2016, pp. 2616–2625.

Salimans, Tim, Kingma, Diederik, Welling, Max. "Markov chain monte carlo and variational inference: Bridging the gap". *International Conference on Machine Learning*. 2015, pp. 1218–1226.

Tucker, George, Lawson, Dieterich, Gu, Shixiang, Maddison, Chris J. "Doubly reparameterized gradient estimators for monte carlo objectives". *arXiv preprint arXiv:1810.04152* (2018).

Wang, Tongzhou, Wu, Yi, Moore, Dave, Russell, Stuart J. "Meta-learning MCMC proposals". *Advances in Neural Information Processing Systems*. 2018, pp. 4146–4156.

Wolf, Christopher, Karl, Maximilian, Smagt, Patrick van der. "Variational inference with hamiltonian monte carlo". *arXiv preprint arXiv:1609.08203* (2016).

Wood, Frank, van de Meent, Jan Willem, Mansinghka, Vikash. "A new approach to probabilistic programming inference". *Artificial Intelligence and Statistics*. 2014, pp. 1024–1032.

Wu, Hao, Zimmermann, Heiko, Sennesh, Eli, Le, Tuan Anh, Meent, Jan-Willem van de. "Amortized Population Gibbs Samplers with Neural Sufficient Statistics". *arXiv preprint arXiv:1911.01382* (2019).