# OpenReview forum: "Nested Variational Inference"
_NeurIPS.cc/2021/Conference — NeurIPS 2021 Poster_

### Official Review · Reviewer_Ae4v · 2021-07-15

**Rating:** 6
**Confidence:** 3

**Summary:**

The paper describes a framework for learning and inference in probabilistic models with structured latent spaces. It operates in the tradition of IWAE and VSMC-like methods that use importance sampling to tighten a variational bound (or equivalently, uses a variational bound as an objective for learning importance sampler proposals). Unlike methods that optimize a single global bound, NVI is defined in terms of local pairwise divergences between a sequence of target distributions, which acts to decouple the stochastic optimization, reducing gradient variance and potentially offering a strategy for parallel inference.

**Limitations And Societal Impact:**

There are no direct societal impact concerns for this work, since the contribution is purely technical and not focused on a particular application.

I do wish there were a bit more discussion of limitations. Have we finally solved probabilistic inference? Is this framework appropriate for all problems? (assuming not, then why?) What improvements or future directions would be worth exploring?

**Main Review:**

I struggled with this paper. On the one hand, I think it likely contains an interesting and valuable technical contribution. On the other hand, the experiments are not particularly compelling, and while that's not a dealbreaker if the idea itself is obviously worthwhile *a priori* (as I sense that the authors understand it to be), I don't think the paper as written is effective at communicating that understanding.

Starting with the abstract and intro, I didn't feel that they conveyed a very clear motivation for the work, beyond 'combine nested importance sampling and variational inference', which is not by itself a goal that anyone knew they had. What useful properties do each of those ideas (NIS and VI) brings to the table, and what do we hope to get by combining them? What kind of tasks, that existing methods are poorly suited for, do we hope to be able to solve? (inference in temporal or spatiotemporal models? multimodal posteriors? high-dimensional state spaces? fitting really deep VAEs? all of the above?) Who should consider using this method? There is a lot of technical detail to get through in this paper, and the introduction could do more to give the reader a sense of why it will be worth it.

I was perplexed by the choice of background material to focus on. The contribution is framed as VI + nested importance sampling, and the paper provides a nice intro to stochastic VI, but no clear definition of nested importance sampling beyond referring to Naesseth et al. 2015/2019. This seems backwards to me. The VAE paper (Kingma and Welling 2013) has 14870 Google Scholar citations, while Naesseth et al. 2015 has just 68; I would suggest that most readers (myself included) are much more likely to need an intro to the latter than the former.

The technical presentation itself is rather abstract and not straightforward to follow. There is reference to an 'NVI objective', but not a clear description of what this objective is --- is it intended to be eqn (6)? I would have expected that an importance sampling objective would at some point ground out to a procedure involving weighted samples, but the concept of weights isn't even introduced until the next section. It also wasn't clear to me how the approach relates to the concept of nested sampling, which I understand from the Naesseth papers to refer to using an 'inner' sampler to generate proposals for an 'outer' sampler. Could this work have been developed in a simpler formalism, like SMC, or is the sampler-within-a-sampler aspect crucial?

Overall I think the presentation tries too hard to be general, at the expense of concreteness and intuition. It might help to frame things in terms of a specific running example, like the multimodal annealing setup from section 4.1 (introducing this earlier would also help give a sense of what kind of problems the approach is hoping to tackle). This would allow you to walk the reader concretely through the steps of applying NVI, and hopefully in the process give some intuition for why it works and where it differs from other methods.

It's not immediately obvious whether the NVI objective is a (stochastic) lower bound on the marginal likelihood, or some similar such property. Is it? If so, I think that'd be a useful argument to include.

I would have liked to see some discussion of implementation, and its ease or lack thereof. As I understand it, the 'juice' of the method comes from estimating gradients using only locally random variables, and an entire page of the paper is about explicitly computing these gradients. This seems to cut against the dominant paradigm of 'just write down the computation graph and let autodiff do its thing'. How straightforward is it to implement all of this in modern DL frameworks? Are there any tricks? You could limit this discussion to the case where all gradients are reparameterizable, since handling discrete variables seems like a somewhat orthogonal (and well-studied) issue.

There's a lot of ink spilled about handling both forward and reverse KL (it's in the abstract even!), but it wasn't clear to me why this is worth focusing on. Does forward KL have any nice properties in this setting that would make up for how much of a pain it is to estimate?

As mentioned above, I didn't find the experiments compelling enough to carry the paper of their own accord. The application to deep generative modeling doesn't contain any quantitative results, or compare to any baselines (except for a reference to RWS in a 'Figure 9' that doesn't seem to have made it into the paper). The multimodal sampling problem is quite toy (though an effective illustration), and the gains over AVO are not super striking. I do appreciate that the experiments seem quite carefully done and are nicely presented, especially figures 3 and 4. In the state-space model experiment, is there any explanation for why optimizing the reverse kernel made things worse?

nits:
line 41: typo 'revserse'
lines 101-103: SMC is listed twice as a special case.

*********************************
Updated: thanks to the authors for your thorough response. I'm convinced that the technical contribution of the paper is solid, and in light of this am raising my score by a point. I encourage the authors to include their clarifications in a revised version of the paper---I think there's likely a much stronger paper here if well explained. The surrogate objective in particular seems worth mentioning and possibly presenting in the supplement (would it make sense to describe it as an extension/generalization of DiCE (https://arxiv.org/abs/1802.05098)?).

**Time Spent Reviewing:**

4

---

> ### Author Response · Authors · 2021-08-11
> **Author Response**
>
>
> Thank you for your thoughtful and constructive comments.
>
>
> ### Motivation / Why VI + NIS
>
> The main goal we have in mind is to provide an inference method which enables us to learn flexible variational proposals for structured models. The high level intuition behind our work can be described as breaking down one difficult inference problem to several smaller (easier) inference problems along the structure of the model. We do this by introducing intermediate densities and defining pairwise VI objectives at each level. The framework of nested importance sampling allows us to reason about the sample generation process at each level independently, which allows us to compute gradients locally. Because of its generality, just like VI, we believe the NVI can be applied to a variety of inference tasks, which we tried to showcase in our experiments. We believe our method can be beneficial for anyone who is interested in inference or training structured deep-generative models.
>
>
> ### Choice of background material
>
> Because one of main novelties of our method is the local gradient computation, we decided to cover the standard VI methods and their gradients explicitly in the background section. However, we agree that having a subsection that more extensively covers NIS in the background would be helpful and we will include it in the revised manuscript.
>
>
> ### Why estimating a forward KL-divergence
>
>
> In the case where we are interested in learning the inference model $q$, the forward KL-divergence allows for an easy handling of discrete variables without the need to resort to continuous relaxations or computing score function gradient estimates, which is why we use a forward KL in the second and third experiment. Due to its mode-covering behaviour it also has the advantage of not underestimate the variance. This is specifically important in the context of importance sampling where one generally wants the variational approximation to be heavier-tailed than the target distribution.
>
> ### What is the NVI objective?
>
> Our variational objective is indeed the objective stated in Equation 6.
>
> ### How does NVI relate to the concept of NIS?
>
> Nested importance sampling only comes into play when we want to compute the ‘gradient’ of Equation 6. Due to space restriction we do not explicitly state the gradient estimators in terms of the nested samplers in the main text. However, we do explain how to compute the gradients, using samples from the previous level of nesting, in Appendix D.4.
>
> ### Could this work have been developed in a simpler formalism like SMC?
> Theoretically, we could formalize the samplers in our experiments as SIR or SMC, but the NIS framework allows us to reason about the sample generation and gradient computation independently at different levels of nesting.
> We will add an algorithm block to the main text to make the computation of the gradients more explicit.
>
>
> ### Relationship to ELBO
>
>
> In general, NVI objective (Eq 6) does not define a bound on the marginal likelihood given that it is defined with a series of f-divergences.
>
> However, if we choose to optimize a KL-divergence, this corresponds to optimizing to a ELBO or EUBO (depending on if we are optimizing a reverse or forward KL-divergence) at the last level of nesting.
> This holds for both, (1) when the final target density $\pi_K(z_K)$ corresponds to some posterior $p(z_K \mid x)$ and $\gamma_K(z_K)=p(z_K, x)$, and (2) when the $z_K$ corresponds to the data $x$ itself (e.g. in the 3rd experiment).
>
> For example, in the case of (1), where $\pi_K(z_K) = p(z_K|x)= p(z_K,x)/p(x)$, we have that
>
> $$\min KL(\pi_{K-1}(z_{K-1})q_K(z_K|z_{K-1})||p(z_K|x)r_{K-1}(z_{K-1}|z_K)) = \min \underbrace {\mathbb E_{\hat \pi_k} \left[ \log \frac{\pi_{K-1}(z_{K-1})q_K(z_K|z_{K-1})}{p(x,z_K)r_{K-1}(z_{K-1}|z_K)} \right]}_{-ELBO}.$$
>
>
> ### Gradients and optimization
>
> The optimization can still be carried out with the usual reverse-mode AD paradigm of modern deep learning frameworks. However, similar to standard score-function based gradient estimators, one can not simply compute the gradient by calling backward on the objective, as the gradient estimator involves analytic manipulations of the expectations, e.g. making use of the reinforce property to simplify the expectation or expanding the gradients of log-normalizing constants to expectation which can be estimated independently. This is the reason why we state the gradient estimators explicitly. We implemented a surrogate objective, which generates the correct computation graph for AD and one can simply call backward on to compute the gradient w.r.t. all parameters jointly. This surrogate objective will be included in our code release.
>
>
> ### Gains over AVO baseline
>
> We believe the improvements of NVI over AVO are substantial. For the non-flow models the best performing NVI-methods (NIVR & NVIR*) achieves over double the ESS (46 -> 97). For the flow-based models we do not compare to AVO but AVO-flow, which is not technically a baseline but a novel combination AVO and flows, that to the best of our knowledge has not been considered before. Moreover, even in this case NVI can significantly improve the ESS for $K=4,6$.
>
>
> ### Optimizing the reverse kernel resulting in a worse performance for the state-space models
>
> Computing ‘full’ gradient updates, including the gradient w.r.t. the parameters of $\check \gamma_k$, requires computing a score-function gradient estimate which yields high variance compared to partial optimization. We hypothesize that this is the reason for the poor performance of the joint optimization scheme.
>
> ### Discussion of limitations & future work
>
> Because of its generality, similar to standard VI, NVI is applicable to a variety of models and tasks. Particularly, we believe it is best suited for models with structured priors. One promising direction for future work is applying NVI to other hierarchical deep generative models. We will add more discussion on the limitations of NVI as well the type of models and applications we believe will benefit from NVI in the revised manuscript.

---

### Official Review · Reviewer_3PA6 · 2021-07-15

**Rating:** 6
**Confidence:** 3

**Summary:**

The author proposes a nested variational inference (NVI) framework that combines nested importance sampling and variational inference. This framework allows the training of the important sampling proposal by minimizing KL divergences at each level of nesting. This also provides us a way to learn intermediate densities, which serves as a heuristics for guiding the importance sampler. Empirically, the author evaluates NVI by (1) sampling from a multimodal distribution with a learned annealing path; (2) learn heuristics for future observations in the state-space model; (3) perform amortized inference in the hierarchical deep generative model.

**Limitations And Societal Impact:**

This paper should not have any negative societal impact as this is mainly a methodology paper.

**Main Review:**

### Original
The NVI framework combines variational inference with the nested importance sampler. The idea of combining importance sampling with variational inference is not new, but the novelty of NVI is that it minimizes the KL divergence at each level of nesting and some of the existing work are special cases of NVI. To the best of my knowledge, this seems to be new. In appendix A, the author also discusses the related work.

### Quality and Clarity
The overall idea of NVI is easy to understand. However, I still think some parts are not explained clearly, and I found them quite confusing when I first read them. I have checked the derivations for the gradient estimate, they seem to be correct. Empirically, the author conducts extensive experiments to confirm the effectiveness of the proposed method.

### Significance
Combining variational inference and importance sampling has recently attracted many research interests. This NVI framework seems to include many existing methods as special cases. Empirically, it seems to outperform the baselines in terms of ESS.

### Questions
1. In the beginning of section 3, I understand why you define a sequence that interpolates between initial $\pi_0$ and target $\pi_K$, and why you define the forward proposal $q_k(z_k|z_{k-1})$. I am not sure I fully understand the role of the reverse kernel $r_k(z_{k-1}|z_k)$. Why do you introduce this? In practice, how do you use this reverse kernel?

2. A follow-up for question 1: why do you want the forward density $\hat{\pi_k}$ to be as close as possible to the reverse density $\check{\pi_k}$ at each level k? What does this objective actually mean? My intuition: If am the one that designs the objective, I will make sure the samples after the proposal $q_k$ should be as close as possible to the intermediate target $\pi_k$, namely the KL divergence between the density of $z_k$ and $\pi_k$. What is the problem of this alternative objective?
 What do you mean
3. I am not sure I understand the meaning of line 127-130. Why it increases the probability of low-weight samples?
4. Why AVO cannot learn the annealing parameter $\beta$? I thought AVO also minimize KL at each level of nesting.
5. Maybe the author can consider adding more background in the appendix to make the paper more self-contained? For example,  the reweight wake-sleep estimator.
**Minor**:
1. Better to add a paragraph to summarize the contributions of this paper.
2. How do you obtain the third equality from the second equality in line 540? Why it becomes $\frac{\partial q_k}{\partial \hat{\phi_k}}$?
3. Maybe consider adding an overall algorithm for easier understanding.

---
I appreciate the author's response. They managed to clarify most of my concerns, especially about the KL objective. Hence, I raise my score.

**Time Spent Reviewing:**

5

---

> ### Author Response · Authors · 2021-08-10
> **Author Response**
>
> Thank you for your thoughtful and constructive comments.
>
> ### Clarity
>
> We will try to improve the overall writing and presentation in the revised manuscript. We would appreciate it if you could elaborate on the parts that lack clarity.
>
> ### Question regarding objective and the role of the reverse kernel
>
> The intuition to make the samples $z_k$ of the proposal as similar as possible to samples from the target density can be formalized as minimize a divergence $KL(\tilde q_k | \pi_k)$ between these two densities, where $\tilde q_k := \int_{z_1, \ldots z_{k-1}} q_1(z_1) q_2(z_2 | z_1) \ldots q_{k}(z_k | z_{k-1}) dz_1\ldots dz_{k-1}$ denotes the marginal density of samples $z_k$. Unfortunately, this marginal is generally intractable.
>
>
> An alternative approach is to define a joint target density $\check \pi_{k}$, consisting of the target density $\pi_k$ and a reverse kernel $r_{k-1}$, and a joint proposal density, consisting of the last target density $\pi_{k-1}$ (in the case of NVI, or the 'forward' marginal of the last step in case of AVO) and the forward kernel $q_{k}$, on the ‘extended’ space $Z_{k-1} \times Z_k$. Minimizing the KL-divergence between these joint densities ensures that samples $(z_{k-1}, z_k)$ from the proposal $\hat \pi_{k}$ are approximately distributed according to the joint target density $\check \pi_{k}$. However, since $\int_{z_{k-1}} \check \pi_{k}(z_{k-1}, z_k)\ dz_{k-1}= \pi_{k}(z_k)$, this also means that $z_k$ is approximately distributed according to the actual target density of interest $\pi_{k}$.
>
>
>
> In the case where $z_K$ is the same as data $x$, we alternatively can think about it in the context of variational autoencoders, where $q$ can be thought of as the generative model while the $r$ as the inference model.
>
>
> ### Intuition on why the estimator for the forward KL-divergence is insatiable
>
> The gradient w.r.t. the parameters of the reverse kernel takes the following form, $- \mathbb E_{\check \pi_k} \left[ \log v_k \ \frac{\partial} {\partial \check \phi_{k-1} } \log r_{k-1} \big(z_{k-1} | z_{k} ; \check \phi_{k-1} \big) \right]$.
>
> Following this gradient for samples with a high weight $v_k > 1$ results in a larger step in direction $-\frac{\partial} {\partial \check \phi_{k-1} } \log r_{k-1} \big(z_{k-1} | z_{k} ; \check \phi_{k-1} \big)$, which results in a decrease of the log-probability of the sample.
>
> Following the gradient for samples with a low weight $v_k < 1$ results in a step in direction $\frac{\partial} {\partial \check \phi_{k-1} } \log r_{k-1} \big(z_{k-1} | z_{k} ; \check \phi_{k-1} \big)$, which results in an increase of the log-probability of the sample.
>
>
> ### Learning annealing parameters in AVO
>
>
> AVO can technically also learn the annealing path, though the authors didn’t perform this additional optimization in the original paper. Nevertheless, NVI and AVO will learn the annealing path in different ways. While AVO could be modified to learn annealing parameters by taking gradients steps w.r.t. the parameters of the target density (which might result in pulling the intermediate densities $\pi_{k}$ towards the initial proposal), NVI learns the annealing parameters by computing gradient updates w.r.t. the parameters of the target density and the proposal density, which is part of the ‘forward’ density. The latter is possible because, unlike AVO, NVI does not propose from the intractable marginal of the previous forward kernels but the target density of the last step, which can be evaluated up to a normalizing constant.
>
>
> ### Additional background material
>
> Thank you for your suggestion. We agree and will add more discussion on reweighted wake-sleep and nested importance sampling to the appendix in the revised manuscript.
>
>
> ### Explicitly list the contributions of this paper
>
>
> Thank you for this suggestion. We will add a paragraph on our contributions in the revised manuscript.
>
>
>
> ### Equality from in the appendix
>
> If we expand the weight $v_k = \frac{\pi_k(z_k;\ \theta_k) r_{k-1}(z_{k-1} | z_k;\ \check \phi_{k-1})}{\pi_{k-1}(z_{k-1};\ \theta_{k-1}) q_{k}(z_{k} | z_{k-1};\ \hat \phi_k)}$ we can see that only the forward kernel $q_k$, which appears in the denominator, depends on parameters $\hat \phi_k$. Hence the partial derivative is $\frac{\partial}{\partial \hat \phi_k} v_k = - \frac{\partial}{\partial \hat \phi_k} q_k$.
>
>
>
> ### Adding an NVI algorithm block
>
>
> Thank you for this suggestion. We agree and will include an algorithm block in the revised manuscript.

---

> ### Author Response · Authors · 2021-08-30
> **Thank you for updating your review**
>
> We are glad to hear that we were able to address most of your concerns. If there are still concerns / open questions, we would be happy to hear and discuss them.

---

### Official Review · Reviewer_FQ8n · 2021-07-15

**Rating:** 7
**Confidence:** 2

**Summary:**

This paper proposes an inference procedure that combines ideas from variational and nested inference, resulting in a sequence of forward and reverse proposal distributions that are optimized with local objectives that encourage the forward and backward edge marginals to be similar. This allows the objective and therefore gradients to decouple. The addition of resampling, afforded by the connection to nested inference, additionally improves sample diversity.

**Limitations And Societal Impact:**

I would like to see speed comparisons between NVI and baselines where appropriate, as that could be a limiting factor in its applicability. For example in the HMM, how fast would exact inference be relative to NVI? This also causes me to be concerned about whether this technique can scale to models with larger state spaces, such as HMMs with thousands of states.

**Main Review:**

*Originality*
* This work proposes a novel combination of prior techniques, namely the local marginal objectives from AVO and the addition of resampling for intermediate distributions.

*Quality*
* The submission is technically sound, and its claims are well-supported by the experimental results. Experiment 1, multimodal sampling, demonstrates quantitative improvements over flow and AVO baselines, both in accuracy as well as sample diversity. Experiment 2, on posterior inference in a small HMM, demonstrates NVI’s use in a problem where intermediate densities are not of constant dimension. Experiment 3 uses NVI on a mixture model on (fashion) MNIST, and contains a favorable qualitative comparison to the same model trained with RWS.

*Clarity*
* The submission is clearly written, although I am hopeful the presentation can be greatly simplified. I would also have liked to see discussion of / relation to AVO earlier than the experiments section.
* The notation might be a bit redundant, with different variable names for (edge) marginals, unnormalized marginals, and proposal distributions. In addition to the different directions, it took a while to get used to the notation. While it does not detract from the paper, any simplifications would be welcome.
* In addition to the appendix, I believe an expert reader could reproduce the results.
* Will the in-text derivation in lines 116-118 be replaced by equations given more space?

*Significance*
* I found the results convincing, although I am not familiar with the SMC literature.
Additionally, the experiments were performed on small data, which leads to concerns regarding scale.

[AVO] Huang, Chin-Wei, Tan, Shawn, Lacoste, Alexandre, Courville, Aaron C. “Improving explorability in variational inference with annealed variational objectives”. Advances in Neural Information Processing Systems. 2018, pp. 9701–9711.


**Time Spent Reviewing:**

4

---

> ### Author Response · Authors · 2021-08-10
> **Author Response**
>
> Thank you for your thoughtful and constructive comments.
>
> ### Presentation
>
> We will try to simplify our presentation and communicate our key contributions more clearly in the updated manuscript.
>
> ### Notation
>
> This paper is indeed notation-heavy. While we’re happy to revisit our notation and see if it can be simplified, we believe that we will still need notation to distinguish between the different density objects (marginal/joint, normalized/unnormalized) to precisely describe our method without requiring extensive space.
>
> We have included the full derivation of the in-text derivations in section 3.2. in Appendix D3.3. Though without the previous space constraints, we will try to include some of the steps described in 116-118 into the main text.
>
>
>
> ### Scalability to larger datasets and speed comparisons
>
>
> As we explained at the end of Section 3.3, NVI requires less memory in the backward pass compared to other objectives for hierarchical models and hence should have a speed advantage in this regard. Therefore, we have no reason to believe that NVI could not be applied to larger datasets. Furthermore, note that our last experiment is performed on the MNIST & Fashion-MNIST images (It takes ~6 hours to train a BGMM-VAE for 50k iteration with a 1080 TI GPU). Regarding the HMM model, a forward pass in NVI entails the same computational cost as in comparable methods. However, as mentioned above, NVI has a memory advantage and potential speed advantage due to the local gradient computation.

---

### Official Review · Reviewer_nv8j · 2021-07-18

**Rating:** 7
**Confidence:** 3

**Summary:**

In this paper, authors present Nested Variational Inference (NVI)--a framework to create increasingly better variational approximations. NVI allows for several existing sampling techniques to be employed in a single framework and is well supported by empirical results.

**Limitations And Societal Impact:**

The proposed methods are still pretty involved and I believe it might need more work for these advanced inference techniques to find there way into every practitioners toolbox.

**Main Review:**

Recent years have seen several papers attempting to combing the Monte Carlo sampling methods and variational inference. I think this paper does a decent job of putting several of those work in a nice single framework. I found the paper to be very well written and easy to follow for the most part. The experiments and carefully carried out and visualizations are very informative. Overall, this is one of the better papers I have reviewed this season--a clear accept.

**Time Spent Reviewing:**

2-3 hours

---

> ### Author Response · Authors · 2021-08-10
> **Author Response**
>
> Thank you for your review. We are delighted by your positive comments about our work!
>
> ### Limitations
>
> It is true that these methods are more involved compared to standard VI methods, and therefore are more difficult to use for practitioners. However, due to their flexibility and advantages in structured models, we believe they are worth investigating and will be valuable to the broader ML community.

---

### Author Response · Authors · 2021-08-30
**Dear Reviewers, Have we addressed your concerns?**

Dear Reviewers,

Have we addressed your concerns? We would be grateful for the opportunity to discuss any remaining issues before the end of discussion period.

Best regards

---

### Decision · Program_Chairs · 2021-09-28

**Decision:**

Accept (Poster)

**Comment:**

This paper learns proposals for a sequence of importance samplers by minimizing a KL divergence for each step (or level) of the sequence. The reviewers consider the work represents a solid and novel technical contribution and is above the acceptance bar for NeurIPS.

The authors are encouraged to improve the presentation and clarity by incorporating the feedback from the reviewers FQ8n, Ae4v, and 3PA6. In particular, consider adding a paragraph/list with contributions, an algorithm box, adding NIS to the background section, and discussing the limitations of NVI. The paper would also benefit from including other technical clarifications, such as the relationship to the ELBO or the intuition on why the forward KL estimator is insatiable.

**Consistency Experiment:**

NeurIPS has a long history of experimentation. In 2014, NeurIPS ran an experiment in which 10% of submissions were reviewed by two independent committees to quantify the randomness in the review process. This year, we repeated a variant of this experiment to see how the quality of the review process has changed over time.  This paper was part of the experiment and was therefore assigned to two committees (consisting of reviewers, an Area Chair, and a Senior Area Chair) that reached independent decisions.  If both committees made the same recommendation, this recommendation was followed. If a single committee recommended acceptance, the paper was accepted (with the exception of a few cases in which the other committee identified what we considered a fatal flaw, e.g., an error in a key result).

This copy’s committee reached the following decision: **Accept (Poster)**

The other committee assigned to the paper recommended **Reject**.  You can find the other set of reviews, along with any follow up discussion with the authors here:
https://openreview.net/forum?id=i2vd6-7bgBi